# Coordinating power sector climate transitions under policy uncertainty

Fikri Kucuksayacigil [1], Zhenhua Zhang [2] & Michael R. Davidson [1,2] ✉

Effective climate policy requires coordination among political jurisdictions with large differences in institutional make-up and interest group structures. We model the effects of realistic coordination barriers and policy heterogeneity on optimal low-carbon electricity pathways in the western United States using a capacity expansion model with operational and coordination mechanism details. We estimate the aggregate costs of fully coordinated power systems (ignoring extreme conditions) to be just a few percent lower than baseline (i.e., current) levels, noting that state-level differences can be an order of magnitude larger. Key infrastructure of a decarbonized power system that facilitates trade such as larger transmission networks is enhanced under multiple complementary coordinating policies, while select location-dependent investments such as energy storage and renewable energy respond more to the degree of market coordination. Degrees of regional coordination have limited impacts on firm low-carbon capacities, whose deployment is driven more by policy stringency, cost and availability assumptions. Climate and energy models should consider the political feasibility of different levels of market and policy coordination when evaluating optimal policy pathways.

Effective climate policy requires coordination among political jurisdictions with large differences in institutional make-up and interest group structures[1,2]. Economy-wide modeling with policy heterogeneity has long shown decreased social welfare and increased policy costs compared to scenarios of policy coordination such as uniform carbon pricing[3,4]. These costs may be partially mitigated in the presence of other forms of coordination—namely, electricity trade[5]—indicating the potential for substituting one type of politically difficult coordination mechanism with another that is more palatable to achieve similar sustainability objectives.

Coordination is particularly salient in low-carbon power systems reliant on large amounts of variable renewable energy (VRE)[6–8], as multiple jurisdictions ranging from balancing authorities to transmission planning regions working together can help address VRE's challenges of spatial and temporal variability by effectively expanding the geographic scope of power systems[9–11]. Yet, power systems coordination is often challenging to implement due to persistent stakeholder preferences for local autonomy and protection of incumbent assets'

value, as well as institutional barriers such as divergent regulatory requirements and policy goals[12–14]. These features are also not well captured in traditional power systems models, which assume a single central coordinator (downplaying frictions of trade) and generally homogeneous energy policy scenarios[9,15,16].

An advanced form of multi-jurisdictional power systems coordination is a Regional Transmission Organization (RTO) or Independent System Operator (ISO) (we use these terms interchangeably): a single entity coordinating centralized day-ahead and real-time dispatch, transmission planning, operating reserves management, and resource adequacy requirements[9,17–20]. In the U.S., RTOs were preceded by lower coordination power pools, and over time, the benefits of transmission coordination and reliability led to the standardization of RTOs[21–24]. To meet decarbonization goals, RTOs can help reduce curtailment and costs by facilitating the trade and integration of renewable energy over long distances[25–28].

In the absence of an RTO, weaker forms of coordination have emerged. In the Western Electricity Coordinating Council (WECC)—the

[1]School of Global Policy and Strategy, University of California San Diego, 9500 Gilman Dr., San Diego 92093 CA, USA. [2]Department of Mechanical and Aerospace Engineering, University of California San Diego, 9500 Gilman Dr., San Diego 92093 CA, USA. ✉e-mail: mrdavidson@ucsd.edu

synchronous grid region encompassing portions of the western U.S., Canada, and Mexico—several regionalization-lite arrangements exist: the Western Energy Imbalance Market (W-EIM) of the California Independent System Operator (CAISO), the Western Energy Imbalance Service (WEIS) of Southwest Power Pool (SPP), and the Western Resource Adequacy Program. The W-EIM is a five-minute real-time market that preserves the autonomy of balancing authorities but lacks transmission planning, resource adequacy, and forward scheduling functions[17]. WEIS provides a similar service and is competing for utilities to join[29], though there exist crucial differences related to management, customer billing, and contract arrangements[30]. While utilities have the most discretion to join the imbalance markets, transforming W-EIM or WEIS into an RTO will require the participation of states, whose political preferences and interest groups are not aligned on crucial energy policy questions[31].

We develop a capacity expansion model with disaggregated coordination functions applied to the western U.S.[2], which represents an important case where coordination is both advantageous and difficult to achieve. Renewable energy resources and electricity demands are heterogeneous and distributed across large distances, generating benefits for co-optimization of VRE utilization across multiple jurisdictions[17]. Nevertheless, the region's power sector is bifurcated through divergent state and utility preferences toward regional grids, institutional conflicts between federal and state governments inhibiting coordination, and varied energy and climate policy objectives[31,32].

In this study, we model the effects of realistic coordination barriers and policy heterogeneity in the electricity sector on optimal low-carbon energy pathways. Specifically, we seek to answer the following questions: (1) Which aspects of power systems coordination are most essential to ensure cost-effective decarbonization? (2) What decisions by individual jurisdictions are robust to uncertainties in the degrees of policy homogeneity and regional coordination? Our contributions are several-fold. First, we estimate the aggregate costs of fully coordinated power systems to be just a few percent lower than baseline (i.e., current) levels of coordination, noting that state-level differences can be an order of magnitude larger. Second, we isolate the impacts of specific coordination mechanisms, revealing that market coordination has a larger impact than long-term planning coordination on key deployments such as energy storage and VRE. Third, key features of a decarbonized power system that facilitate trade such as larger transmission networks are enhanced under multiple types of coordination, indicating the presence of a complementary suite rather than a set of substitutable policy levers for coordination.

## Results
### Modeling inter-jurisdictional coordination
In contrast to issues of technical inflexibility (operations, interconnections), coordination inflexibility has received less attention from power system modelers. When inter-jurisdictional frictions are modeled, they may include synthetic transaction cost barriers known as hurdle rates that increase the cost of trade between two non-coordinating zones[18,25,33–35]. Limiting the sharing of reserve generation between non-coordinating zones can also reflect the primary demand balance responsibilities of balancing authorities[9,16,35,36]. Setting resource adequacy requirements at the zonal—instead of grid-wide—level reflects the role of individual sub-national entities and utilities in maintaining reliability[16–18]. Enhanced market coordination has been evaluated in moving from zonal price formation, such as the European single market, to more granular nodal pricing[37,38]; integration of redispatch function into the current market[39]; the synchronization of the Baltic grid with that of continental Europe[40]; and coordination of national grids in northwest Europe[41]. With respect to policies, prior studies have considered state-level policy targets and renewable energy certificate (REC) trading, which adds computational burdens[15,16].

Parametric and structural uncertainties are key features of energy system models[42]. For long-term deep uncertainty, policy scenario analysis can generate robust decisions that behave well under different plausible futures[43] and calculate the value of flexibility in designing capital-intensive systems[44]. Robust decision-making frameworks can also reveal the design with the least regrets based on simulation of different uncertain realizations including institutional uncertainties[45].

The efficiency of electricity markets has been evaluated by econometrics models retrospectively based on empirical data. The benefit of market and different dispatch schemes on generation output and cost were examined using a difference-in-difference analysis considering the staggered transition to markets[46]. The benefit of centralized dispatch in the context of PJM's expansion to areas previously trading electricity with bilateral contracts has been shown to be large[47]. These results follow from a long literature noting the prospective benefits of greater market integration[48–51]. Econometric techniques are well-suited when there is abundant data and the research focus can be a well-specified set of historical events. In order to assess forward-looking outcomes, e.g., for 100% clean energy by mid-century, then a different class of models is needed, such as optimizations which can consider dynamics of changing markets, policies and infrastructures while respecting technical constraints.

We build a power system capacity expansion model with hourly operational details tailored to addressing questions of coordination. Capacity decisions pertain to resources (generation resources excluding storage and transmission), storage units, and transmission lines, and operational decisions include hourly dispatch and flow on transmission lines. The model is run for 12 demand zones in WECC, excluding Canada and Mexico (Fig. 1, this model does not consider boundaries and corresponding frictions between balancing authorities within the same modeling zone). We consider five dimensions of market coordination—energy trade, operating reserves, resource adequacy, transmission planning coordination, and generation planning coordination—and construct five scenarios of regional markets: (i) Incomplete Coordination (the lowest level of market coordination), (ii) business-as-usual (BAU; zones outside of California's existing market, CAISO, have low coordination), (iii) Expanded EIM (frictionless energy balancing is expanded to all of WECC), (iv) Regional Market (single

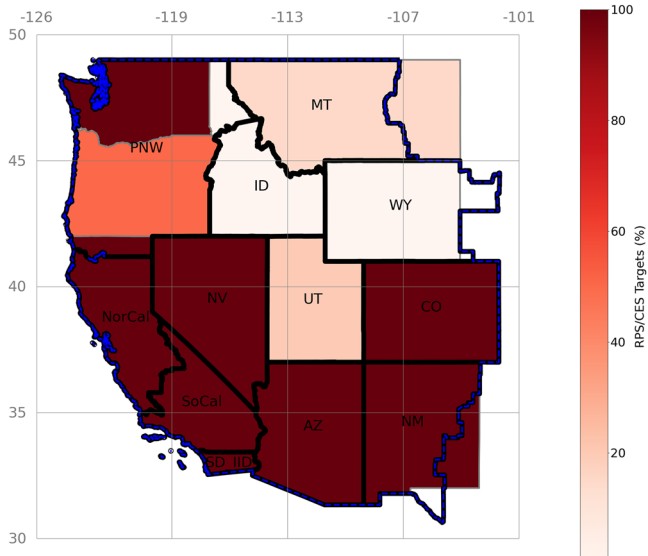

**Fig. 1 | Modeling zones (black,[76]) within WECC (dashed blue,[76]) and state borders (gray,[77]) considered in our study.** State borders do not necessarily overlap with modeling zones or WECC area due to differences in utility service areas. The scale indicates state-level CES or RPS targets extrapolated to 2050. Sources: U.S. EPA, U.S. Census Bureau.

coordinator), and (v) Full Coordination (the highest level of market coordination). We also consider heterogeneous state clean energy goals: (i) current state targets in terms of stringency and technology coverage by 2050 (Fig. 1), equivalent to an average 68% region-wide clean energy standard (CES) (Continued State Policy) and 54% reduction in CO2 emissions compared to 2021, and (ii) regional 100% CES goal by 2050 (Regional 100% CES) achieving 98% CO2 emissions reductions considering some small residual emissions from partial carbon capture rates. Continued State Policy scenario (abbreviated as State Policy henceforth) is more optimistic than the current clean energy goals of the states (for more information, see Methods and Supplementary Information). For a quick review and description of five regionalization scenarios and two clean energy policy scenarios, see Table 1, Table 2, and Table 3.

### Low-carbon capacity expansion to meet strengthened policy

Enhancing regional coordination reduces costs across each of the policy scenarios (for reference, cost of State Policy + BAU, the second bar in the left panel of Fig. 2, is $36.73B/yr and cost of Regional 100% CES + BAU, the first bar in the right panel, is $41.98B/yr). The Regional Market brings about approximately $850M/yr and $370M/yr benefits under State Policy and Regional 100% CES, respectively (Fig. 2). The benefits of weaker coordination in the form of the Expanded EIM are in the range of $330M/yr - $610M/yr. These cost savings are approximately 15% of the policy costs of moving from State Policy to Regional 100% CES, which are around $5B/yr, and a few percent of total power system costs in 2050. High policy attainment costs moving from State Policy to Regional 100% CES derive from the increased reliance on gas with carbon capture and storage (CCS), in line with prior literature showing a steep increase in the last stages of power sector decarbonization to provide firm capacity[6,52] (we define firm resources as biopower, non-CCS coal, CCS coal, geothermal, non-CCS gas, CCS gas, hydropower, nuclear, and other small-sized plants such as municipal waste). Low-cost scenarios of solar, wind, and battery storage result in a substantial decline in total costs relative to the reference cost case,

though comparable savings relative to BAU with the same cost assumptions.

For State Policy, results are obtained with 75% in-state fractions for renewable portfolio standards (RPS) or CES requirements (depending on the state) for Incomplete Coordination, BAU, Expanded EIM, and Regional Market scenarios, i.e., 75% of the clean generation to meet the state's policy should be delivered from in-state resources, following[53]. With enhanced coordination in generation planning (Full Coordination scenario), the costs of achieving the same policies are reduced by a further $17M/yr. Cost savings of coordination in transmission alone are $125M/yr - $2.23B/yr. When including all five dimensions of inter-jurisdictional coordination, the costs of achieving State Policy are reduced by $3.25B/yr.

Regional 100% CES policy requires substantial capacity additions of VRE and storage, much larger than changes associated with different levels of market coordination (Fig. 3). However, regardless of the clean energy policy scenario, markets have a large impact on resource locations, e.g., shifting solar investment centers to states with high clean energy targets and lower build costs (Fig. 4). Firm capacities (both carbon-emitting and low-carbon) do not substantially change in aggregate or in deployment location under different levels of policy and market coordination (Figs. 3 and 4). Firm capacity deployment is instead driven by policy stringency, cost and availability assumptions. Under State Policy, which does not achieve net-zero emissions (See Supplementary Fig. 19), non-CCS gas is still prevalent and is supplemented by a small portion of non-CCS coal and other carbon-emitting resources. Regional 100% CES retires all non-CCS gas and coal units and leads to the installation of CCS gas and biopower (a small amount of GHG emissions in this scenario due to partial capture rates of newly installed CCS gas units). While natural gas combined cycle resources are being built by the model in State Policy scenarios, biopower, geothermal and CCS gas are being built by the model in Regional 100% CES scenarios. Enhanced coordination in markets needs more transmission lines, but Regional 100% CES policy reduces this need substantially, with greater deployment from storage units, explored next.

### Transmission and storage complementarities and coordination

Transmission and storage are widely recognized to have a substitution relationship in a low-carbon power system as both can help manage variability in supply and demand[54,55]. However, the political economies of storage and transmission are quite distinct, with transmission generating a wider range of opposition based on land use conflicts, conservation priorities, and nuisance[56,57]. Increased regional coordination leads to larger transmission and lower storage expansion (Fig. 3). Storage capacity increases by 54% in the presence of market and

### Table 1 | Clean energy policies considered in Continued State Policy scenario

| State | RPS | CES |
|---|---|---|
| AZ | 15% by 2025 | 100% by 2050 |
| CA | 60% by 2030 | 100% by 2045 |
| CO | 30% by 2020 | 100% by 2050 |
| ID | - | - |
| MT | 15% by 2015 | - |
| NV | - | 50% by 2030, 100% by 2050 |
| NM | 80% by 2040 | 100% by 2045 |
| OR | 25% by 2025, 50% by 2040 | - |
| UT | 20% by 2025 | - |
| WA | - | 15% by 2020, 100% by 2045 |
| WY | - | - |

### Table 3 | Clean energy policy scenarios

| Name | Targeted RPS and CES values |
|---|---|
| Continued State Policy | Zones meet 2050 targets |
| Regional 100% CES | 100% CES are met region-wide |

### Table 2 | Regionalization scenarios

| Name | Definitions | | | | |
|---|---|---|---|---|---|
| | Hurdle rate cost | Operating reserve | Planning reserve | Tr. coord. | Gen. coord. |
| Incomplete Coord. | Non-zero | Zonal level | Zonal level | Limit | Lower |
| BAU | Non-zero | Zonal level | Zonal level | No Limit | Lower |
| Expanded EIM | Zero | Region-wide | Zonal level | No Limit | Lower |
| Regional Market | Zero | Region-wide | Region-wide | No Limit | Lower |
| Full Coordination | Zero | Region-wide | Region-wide | No Limit | Enhanced |

planning coordination barriers compared to the Full Coordination case, demonstrating its important role in mitigating the deficiencies of multi-jurisdictional power systems.

Transmission expansion decisions can be highly sensitive to state-level clean energy policy assumptions, with implications for near-term planning. Pathways connecting states with low-ambition State Policy (lighter color in Fig. 1) may be expanded if those states are exporters to states with high clean energy targets (darker color in Fig. 1). However, under Regional 100% CES, those same states require more clean power within the state, reducing their exports to neighboring states and thus inter-state transmission capacities (relative to State Policy). States with high-ambition State Policy shift their import dependencies to different pathways when the whole region must meet 100% CES. See Supplementary Figs. 3 and 8 for power flows between regions and for all transmission pathway expansion results.

Transmission builds are enhanced by complementary coordination conditions, notably in markets. Market regionalization is a prerequisite for enhanced transmission planning coordination to yield new investments (Fig. 3). Transmission expansion is around 18–136% of the existing grid capacity in BAU, and around 28–149% in Regional Market. These complementary rather than substitutable levers of coordination are key limitations of bottom-up climate policy.

## Robust planning under policy uncertainty

Robust decisions reflect no-regret investments regardless of policy and other uncertainties (no-regret capacity refers to the capacity that is built regardless of which scenario is realized and thus represents a robust decision by states and utilities to future uncertainties). The existence of both policy and market uncertainties is a roadblock for variable renewable energy resource deployments: states with abundant renewable resources nearly double the no-regrets capacity when policy uncertainty disappears (i.e., change from All to 100% CES in Fig. 5). It is not surprising that low-ambition states have low build-out of renewable resource deployments with policy and market uncertainties ahead. Wind energy is the least sensitive resource in the face of market and policy uncertainties, owing to its more locationally-specific resource quality. On the contrary, investments in solar and battery

change substantially when a more predictable policy is put forward. Across the scenarios, battery capacity in CA also remains high (23–45 GW), in the range of other studies finding that battery capacity in the state should be in the range of 20–100 GW[27].

Resource planning by utilities and governments generally assumes a continuation of current market structures. However, ongoing institutional changes may lead to different market formulations and changes to levels of policy coordination. To analyze the impact of unexpected changes in market policy on low-carbon outcomes, we perform a two-stage optimization. Assuming partial information, we fix investment decisions under the BAU scenario, then run operations assuming a Regional Market. Costs of operating the power system increase by $109M/yr (variable cost accounts for 12% of the total power system cost). The reverse analysis—optimizing investments for Regional Market and operations under BAU—was infeasible due to unmet requirements for zonal resource adequacy.

We perform sensitivity analysis with low-cost solar, wind, and batteries, as well as high-cost gas (Fig. 6). We observe complementaries between solar and storage, and between wind and transmission. When either solar or storage is cheaper, investment in both accelerates, and investment in wind reduces. Cheaper wind leads to deceleration in solar investments, and expansion in power lines substantially increases, in line with other studies[9]. Solar investments are larger in low-cost storage compared to low-cost solar, owing to the strong complementarity of the two resources and the lower end of future storage cost projections (see Supplementary Fig. 20 as complementary to Fig. 6, where we show reference cost case capacity and relative change on top of those capacity values).

We conduct additional sensitivity analysis by running all scenarios with different in-state fractions of RPS/CES goals (default/baseline values are 0% for Full Coordination and 75% for all other market coordination scenarios), operating reserve requirements (default is 3 + 5 heuristic), and hurdle rates (see Methods and Supplementary Information for more details). Decreasing in-state CES/RPS requirements (i.e., increasing REC flexibility) from 100% to 0% leads to approximately $200M in cost savings – about 0.7% of the system cost in the BAU scenario – across all scenarios due to less wind and solar

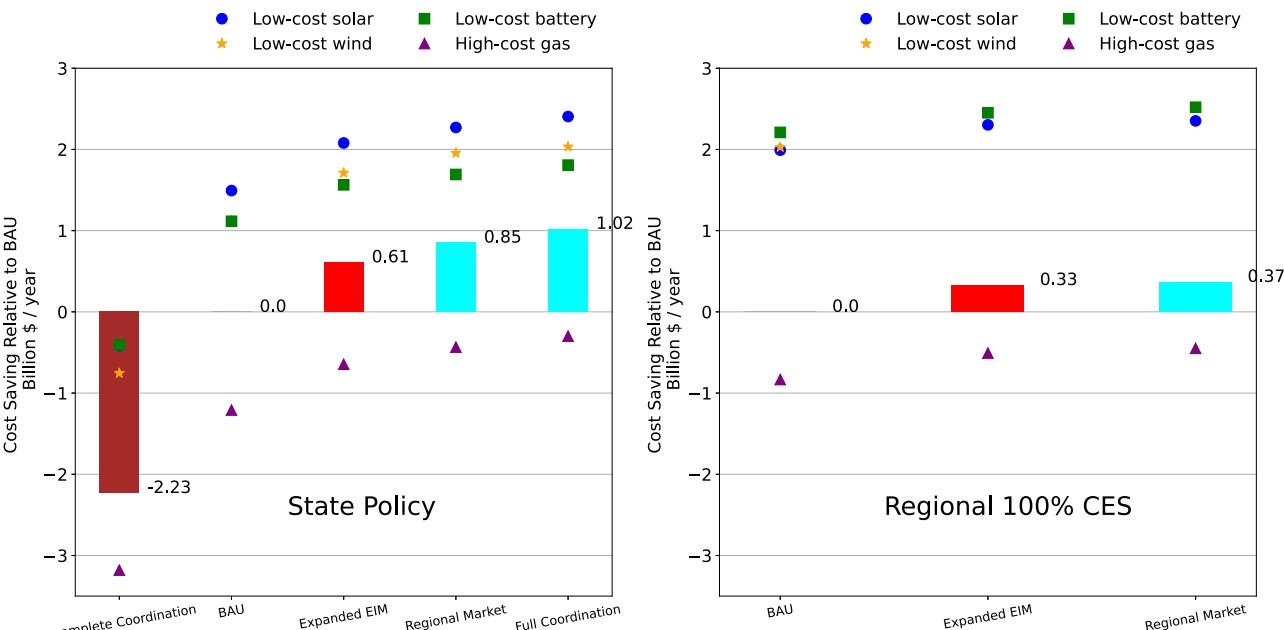

**Fig. 2 | Cost savings of the power system in 2050 under different levels of policy and market coordination for State Policy (left) and Regional 100% CES goals (right).** Bars represent scenarios of regionalization and clean energy policy. Dots

represent sensitivity analyses keeping the model structure the same. For more explanations about scenarios, see the Methods section.

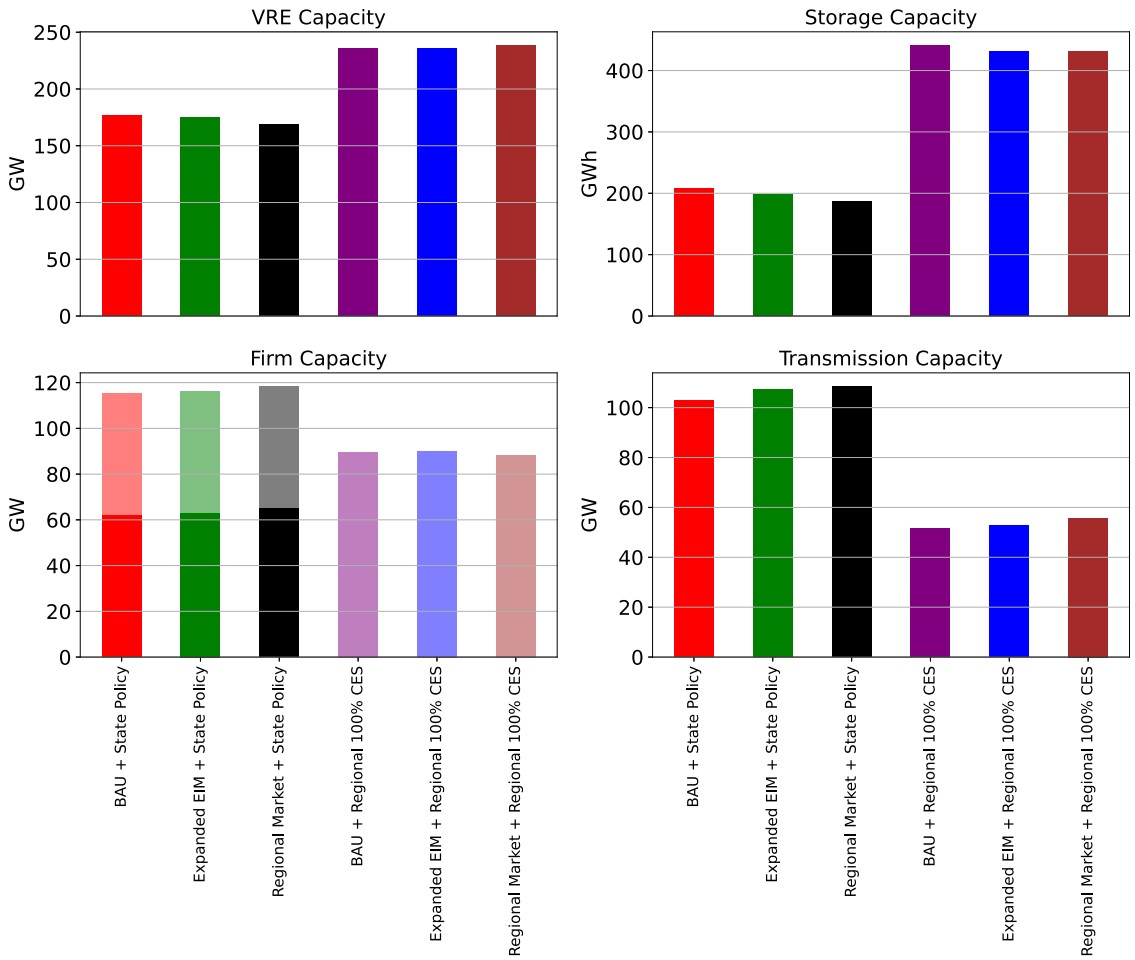

**Fig. 3 | Power system capacities for all scenarios.** Firm capacity includes both carbon-emitting (dark shading) and low-carbon resources (light shading). VRE, storage, firm, and transmission capacities are shown in top-left, top-right, bottom-left, and bottom-right panels, respectively.

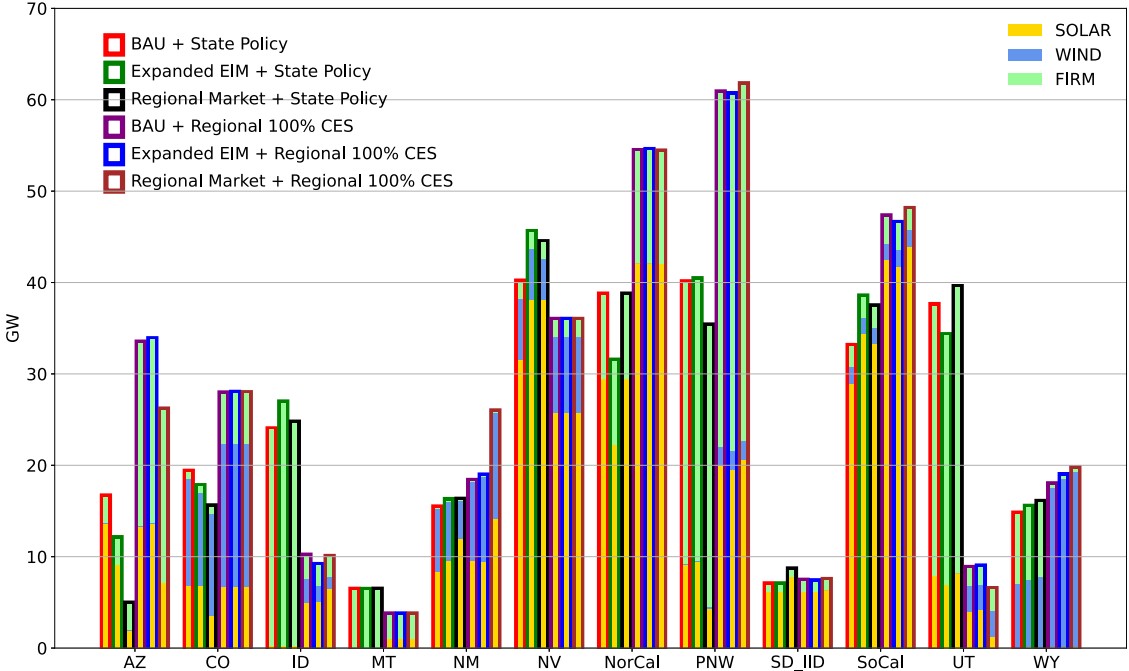

**Fig. 4 | Solar, wind, and firm resource capacities by state in 2050.** Borders indicate scenarios and fill colors indicate resource type.

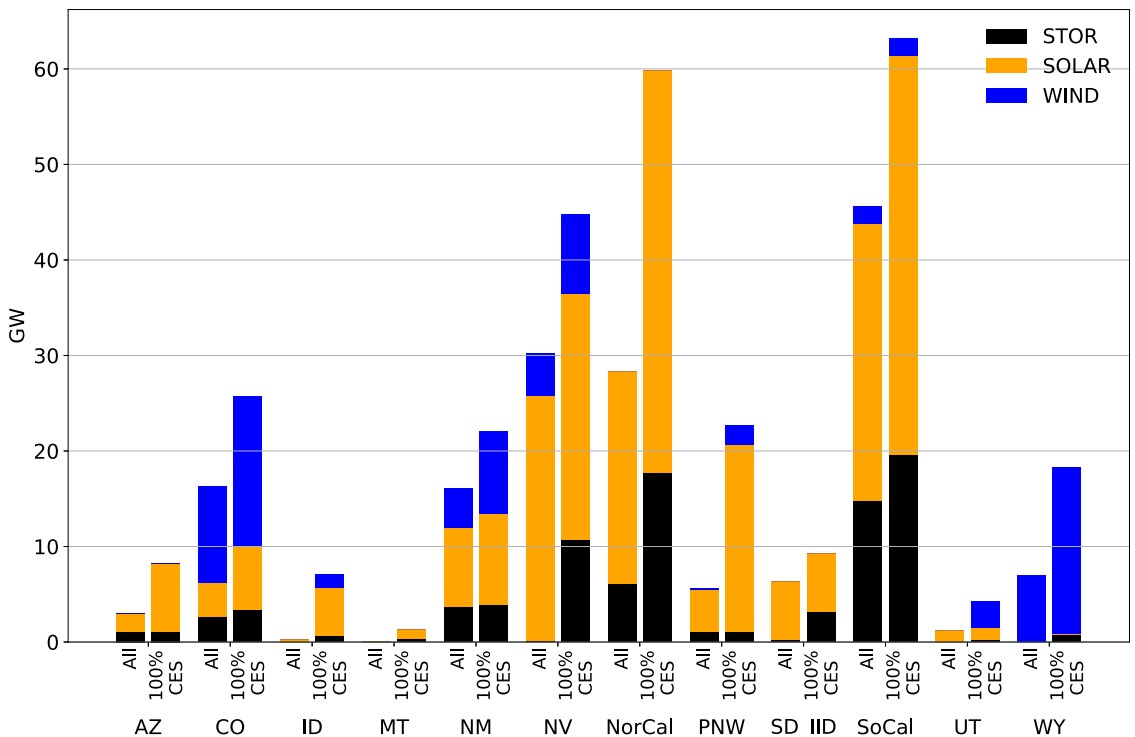

**Fig. 5 | Robust capacities for all scenarios and for only 100% CES scenarios.** All refers to the minimum capacity of a resource across the main six scenarios described in Fig. 3 and Fig. 4 and 100% CES refers to the minimum capacity of a resource across three 100% CES scenarios.

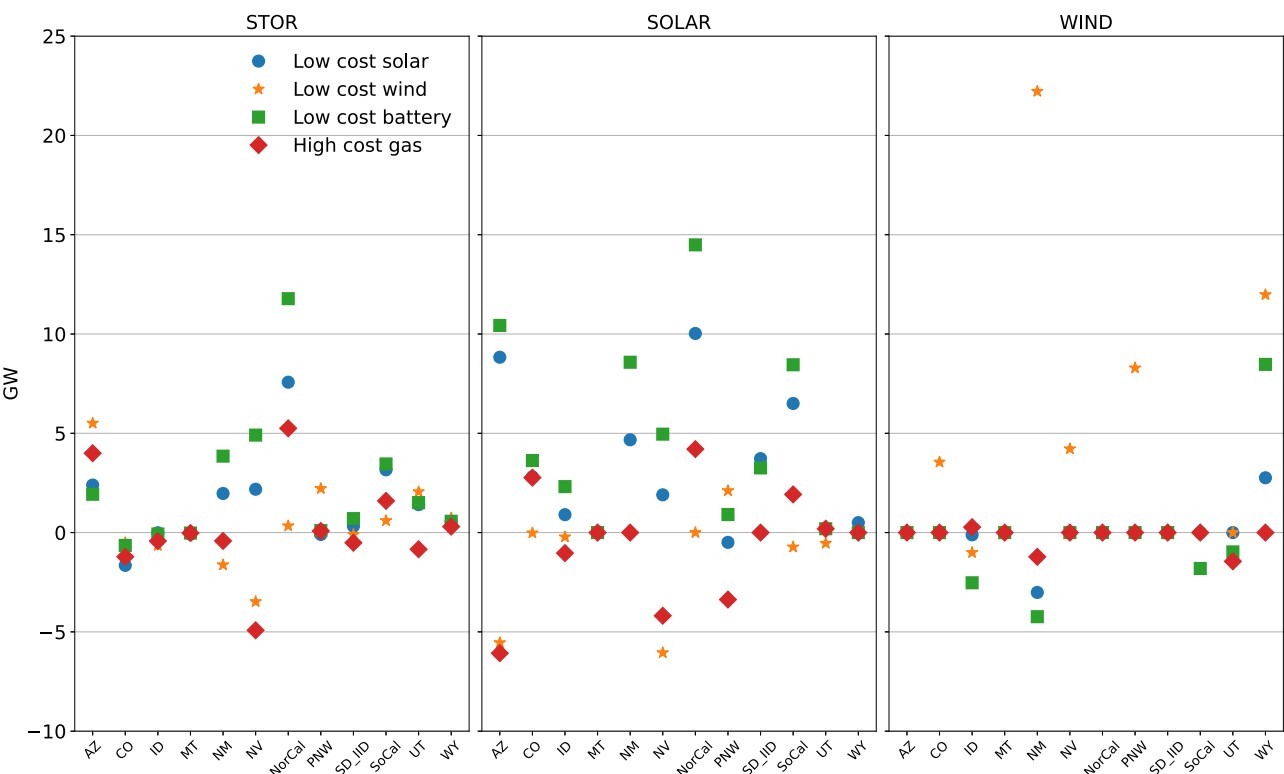

**Fig. 6 | Cost sensitivities around BAU + Regional 100% CES scenario for low cost solar, wind and battery, and high cost gas.** Capacities are shown relative to baseline costs for storage (left), solar (middle), and wind (right).

deployment. Higher operating reserve requirements lead to increases in system costs of less than $90M due to the additional storage deployment. Varying hurdle rate costs can lead to system cost savings of up to $180M – about 0.5% of the system cost in the BAU scenario – driven by the additional solar deployment in low-rate zones and less wind deployment in high-rate zones. As indicated before, the aggregate system costs of fully coordinated power systems remain just a few percent lower than baseline levels, with large region-wide differences.

## Discussion

Power systems coordination (e.g., regionalization of markets, homogeneous policy targets, and generation and transmission planning coordination) reduces system costs and improves the viability of renewable energy, and is, therefore, an important element of climate policy coordination. Ultimately, the decision to coordinate is political. This research is motivated by real-world coordination decisions facing governments and stakeholders and their implications for cost-effective climate mitigation.

Results show that variations in regional power market coordination bring about alterations in system design in terms of resource selection and costs. Enhanced coordination has a larger impact on the distribution of resources and transmission lines compared to the region-wide capacities of these resources. First, increased coordination gives states more flexibility in selecting investment locations to meet their policy goals, particularly VRE and storage. Second, regional coordination necessitates less storage support for solar projects, whose intermittency can be partially managed by inter-state power transfer. However, the degree of regionalization has limited impacts on firm low-carbon capacities, whose deployment is driven more by policy stringency, cost and availability assumptions rather than network and market coordination. Broad implications for climate policymakers facing uncertainty in coordinating neighbors point to the possibility for region-wide planning of firm low-carbon capacity even under market uncertainty, while storage has an important role in mitigating the deficiencies of low coordination power systems.

Including all levers of coordination increases savings by $3.25B/yr, though this is likely an underestimate. Other modeling studies that incorporate demand forecasting integrated with electric vehicle charging, more advanced resource siting decisions, and enhanced storage options show large benefits even considering a smaller set of coordination features[17,18]. Our results indicate that Expanded EIM's cost saving is around $330M/yr - $610M/yr, while the Western EIM reports that the cost savings to its participants were $1.4B in 2022. The latter benefits are attributed to extreme conditions of that year, such as strong hydro conditions, high gas prices, and increased demand[58], not simulated here.

Our results point to potentially important facets of transmission expansion planning as governments enhance climate ambition. States with low (or no) clean energy targets but high renewable energy potential expand their transmission lines to export power to states with high-ambition targets, but they need less inter-state transmission expansion when they themselves adopt stringent clean energy targets as they keep more solar and wind generation in-state. Coordinating interstate transmission planning and cost allocation are notoriously difficult[12,59]. Therefore, regardless of future regional climate ambition, political factors shaping cost allocation and beneficiaries of transmission projects will be highly consequential. Policy support for the resolution of transmission roadblocks emphasized by the Department of Energy and proposed legislation in Congress must also address market creation[60].

In enhancing ambition from State Policy (approximately 54% CO2 reductions from 2021 level) to Regional 100% CES (98% reduction), states deploy firm low-carbon resources and rely more on solar complemented with storage, decreasing the need for additional transmission expansion. This does not contradict the large body of studies that show greater transmission requirements associated with stringent greenhouse gas (GHG) emissions reduction targets[61–63], but does indicate that the baseline matters. Ref. 26 finds WECC-wide transmission expansions in the range of 30–54% in GW over the existing grid to reduce GHG emissions by 80% by 2050 relative to 1990 levels (our results are 18–149% across market and policy scenarios).

Our study has several limitations. We model coordination through five distinct functions, though in reality, there are many stages through to full regionalization. For instance, instead of joining a full market, some states and utilities may be more prepared to join real-time markets, day-ahead markets, and/or regional resource adequacy programs. It is also possible that states and utilities will choose different configurations—including multiple regionalization entities (such as bifurcated EIMs), a regional market without key states, or where utilities in the same state join different RTOs[17,35]. Our study does not capture all the discrete elements of coordination due to computational tractability and data availability, thereby underestimating the benefits of coordination. Our study does not incorporate transmission network expansion that must take place within states for feasible dispatch. What determines states' RTO participation is an open area of debate: policy diffusion could lead to a domino effect of states choosing regional coordination. Our modeling results rely on sampled weeks (including peak week) from a single year, ignoring potential extreme conditions. Furthermore, we utilize aggregated modeling zones that ignore in-state frictions. For example, all western states have more than one utility, and power transfers between those in-state utilities are subject to friction, ignored here. Higher spatial granularity would come with important computational and practical challenges. Operating reserves are shared over the region when coordination increases, but we do not have deliverability details other than long-term contractual imports and exports. We believe these provide promising directions for future research.

Coordination in power systems is beneficial for reaching decarbonization goals, yet full coordination is politically elusive. We developed a state-level capacity expansion model with operational details and formulations of coordination that enable a granular view of the contributors to cost and clean energy improvements. This approach, and extensions to economy-wide models with a broader range of policy coordination levers[2], will be useful for power system planners and researchers seeking to explore economically efficient and politically realistic solutions to the low-carbon transition.

## Methods

### Data sources

We use PowerGenome (PG)[64] to obtain the majority of input data. PG is an open data platform that extracts raw data from various public sources and constructs datasets of generators, resource profiles, estimated load, fuel costs, and transmission networks. Demand and renewable energy profiles are taken from 2012. We supplement with data from the California Energy Commission[65] and corresponding RESOLVE model input data[66], generation adequacy by Western Flexibility Assessment[67], and several additional sources[27,36,68–70]. For a complete description of input data and their sources, see Supplementary Information.

### Model

We develop a capacity expansion model in Julia/JuMP[71,72], building on notebooks in ref. 73. Gurobi is used as the optimization engine and the model is run on a high-performance cluster. See Supplementary Information for more on the computational setup.

### Decision variables

Decision variables of the optimization model include investments and operations. Investment decisions include new build and retired capacities of generation resources, storage, and transmission lines. Hourly operational decisions include commitment states of thermal units, dispatch, the charge of storage units, non-served energy, flow on transmission lines, and reserves. Generation resources include CCS coal, CCS gas, non-CCS coal, non-CCS, gas, wind (both land-based and offshore), solar, petroleum liquids and other small-sized resources, biopower, nuclear, hydropower, and geothermal.

### Constraints

Investment potentials for new build solar, wind, geothermal, and hydro projects, demand balance for zones, transmission expansion,

maximum/minimum dispatch, reserve operations, time coupling, storage operations, commitment operations, and minimum up/down times for thermal units stand as constraints. The model also includes constraints of RPS and CES requirements and operating and planning reserves.

## Objective function

Objective function minimizes system-wide annual cost. It includes annualized investment and fixed costs of resources, storage units, and transmission paths, variable cost of generators, cost of non-served demand, startup costs of thermal plants, and hurdle rate costs in the Incomplete Coordination and BAU scenario. See Section 3 of Supplementary Information for more details.

## Zonal aggregation

We model eleven U.S. states in the WECC to analyze the effect of western states' optimal actions on California's electricity system's future. WECC includes parts of fourteen U.S. states and some portions of Canada and Mexico. We ignore the small parts of Texas, South Dakota, and Nebraska within WECC. Each balancing authority is accountable for its own supply-demand balancing. We perform zonal aggregation to create 12 demand centers/zones, combining OR and WA and separating CA into three zones. The choice of zones is based on institutional infrastructure, the similarity of load and generation profiles, and the potential for transmission congestion between zones[33].

## Resource clustering

We cluster generator unit commitments to mitigate the computational burden[74]. We implement clustering based on the similarity of operational parameters such as heat rate, resulting in 480 total clusters. For each clustered generator, we assign an integer variable for commitment decisions. For non-thermal generators, we aggregate all resources of the same type in a single resource cluster.

Out of 480 clusters, existing generators represent 105 clusters. Each modeling zone has one cluster to represent a distinct resource type (natural gas, biopower, solar, wind, etc.). Existing solar and wind resources are represented with a single cluster in each zone. New-build resources represent 375 clusters. New-build solar and new-build wind resources have multiple clusters, hence multiple resource profiles can be chosen by the model.

## Time sampling

Our model simulates six representative weeks in the year and simulates the energy system operations for each hour during selected weeks. PG determines representative time periods via zonal aggregation, resource clustering, and time sampling. It uses a k-means clustering algorithm to pick the representative weeks, and includes the peak week during the year to ensure resource adequacy conditions are met. Peak loads are computed separately by zone (in the zonal resource adequacy scenario) or a single region-wide peak load (in the regional resource adequacy scenario). Hourly load data over the sampled six weeks is taken from PowerGenome and includes baseline electrification levels.

## Proposed transmission lines

In addition to the existing inter-zonal transmission network, we identify proposed inter-zonal transmission lines (in total, 16) that have passed some level of feasibility planning stage by relevant grid authorities. Proposed lines are used to parameterize transmission coordination, described next. For a list of proposed transmission projects, see Section 5.13 of Supplementary Information.

## Scenarios

We envision that four substantial types of coordination in the western states will shape the future electricity system: (i) degree of regionalization, (ii) clean energy policy targets and stringency, (iii) transmission planning coordination, and (iv) generation planning coordination. We aggregate (i), (iii), and (iv) to create five market coordination scenarios.

**Degree of regionalization.** We assert that an RTO has, at minimum, three key functions: frictionless electricity trade across multi-jurisdictions, shared operating reserve, and resource adequacy requirements[17,18]. The BAU scenario assumes that existing barriers to coordination persist. Balancing authorities will continue trading electric power with bilateral negotiations, which are subject to jurisdictional frictions. The Expanded EIM scenario envisions that the current EIM of CAISO expands to the entire WECC, and includes energy balancing and reserve sharing, but not resource adequacy sharing. The Regional Market scenario assumes a complete regional market with all three coordination functions.

In Expanded EIM, we assume that the planning reserve is zonally met, but the import and export capabilities of states increase. This scenario is similar to increased coordination under the bilateral market evaluated in ref. 18 and similar to the expanded real-time market investigated in ref. 17. We utilize hurdle rates to represent jurisdictional barriers to energy balancing. It may consist of bilateral trading transaction costs, wheeling charges, administrative transmission tariff charges, bilateral trading margin, and market friction resulting from changes in unit commitment cycle[18]. Using hurdle or wheeling rate costs is common practice among modeling studies to represent non-techno-economic barriers[33]. We apply hurdle rate costs across zones based on the exporting region (see Section 5.14 of Supplementary Information). There is no hurdle rate for trade between California zones.

**Clean energy policy targets and stringency.** The majority of western states have clean energy targets, though collectively they fall short of a long-term commitment to 100% clean energy. The Continued State Policy (abbreviated as State Policy henceforth) scenario is more optimistic than current formal targets in some cases. If major utilities in the state have adopted 100% CES or similar targets, or if the state's other climate goals create a de facto CES, then we adopt 100% CES for that state (Table 1). These are elaborated in Section 7 of the Supplementary Information. We assume that RPS-eligible resources are the same across our modeling zones, which also holds for CES-eligible resources.

The second scenario, Regional 100% CES (including large hydro, nuclear, and CCS), is motivated by the complete decarbonization of the western electricity system, which is equivalent to each state having 100% CES (ref. 18 performs a sensitivity analysis for increased RPS in California).

**Transmission planning coordination.** Our study assumes that transmission can be expanded on top of the existing network along any viable pathway without any maximum capacity limit, a common modeling convention, which we denote Transmission Coordination. Proposed transmission projects reflect current levels of coordination among utilities, states, and communities. In the Limited Transmission Coordination scenario, we allow the model (but do not require it) to expand the existing grid only through the paths and up to the planned capacities of the 16 proposed inter-zonal transmission projects.

We only focus on inter-zonal transmission line expansion in this study. While we omit explicit intra-zonal line expansion, we integrate their costs (e.g., connecting generators to the main grid) in inter-zonal line expansion.

**Generation planning coordination.** With Lower Generation Planning Coordination, we run State Policy with a minimum of 75% in-state renewable energy. We decrease this parameter to 0% to run Enhanced Generation Planning Coordination.

We summarize regionalization scenarios in Table 2 and clean energy policy scenarios in Table 3.

We run regionalization and clean energy policy scenarios combinatorially. Complete set of scenarios include (i) Incomplete Coordination + State Policy, (ii) BAU + State Policy, (iii) Expanded EIM + State Policy, (iv) Regional Market + State Policy, (v) Full Coordination + State Policy, (vi) BAU + Regional 100% CES, (vii) Expanded EIM + Regional 100% CES, and (viii) Regional Market + Regional 100% CES.

**Two-stage optimization.** In the two-stage optimization, we solve our model first assuming scenario X (e.g., BAU markets). Next, we fix the investment decisions (generation, storage, and transmission) obtained from scenario X and then solve for the operations of the system under an alternative scenario Y (e.g., Regional Market). The cost and effects of the uncertainty are measured by the difference between the two-stage result and the single optimization under scenario Y.

**Sensitivities.** We perform cost sensitivities on BAU + Regional 100% CES scenario. We test lower costs of new build solar, wind, and batteries and higher cost of gas units[27,68]. Low-cost sensitivities of solar, wind, and battery use the Advanced Cost Case of the Annual Technology Baseline of NREL[75] (cost values are in 2018 dollars for both reference and sensitivity). We obtain the high-cost gas scenario in two steps. First, we multiply reference capital cost and fixed O&M cost by 1.3, reflecting the difference between NorCal and SoCal reference costs[27]. Second, we replace the reference fuel cost for mountain and pacific natural gas (CCS and non-CCS) with ref. 27's fuel cost for the same areas (they do not indicate this as high-cost gas, but these costs are higher than our fuel costs). For cost data, see generator data files in our repo corresponding to reference cost cases and sensitivity analysis.

We perform sensitivities to evaluate the robustness of results around key regionalization modeling parameters, namely, in-state fractions of RPS/CES goals, operating reserve requirements, and hurdle rates. We run all scenarios given in Fig. 2 with different values for these parameters. A detailed explanation of the sensitivity analysis design is given in Supplementary Information.

## Data availability
All source data, modeling inputs, and generated outputs from this study are available on Zenodo at https://doi.org/10.5281/zenodo.14783689.

## Code availability
The code used for this study, including analysis scripts, is available on GitHub at https://github.com/Power-Lab/NatureComms_Market_Coordination_2025.

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

## Acknowledgements

We acknowledge the assistance of Greg Schivley and Jesse Jenkins in configuring and using PowerGenome. We appreciate the constructive feedback from attendees at the INFORMS annual conference and staff at the California Energy Commission, California Independent System Operator, and California Public Utility Commission who reviewed an earlier version of this work.

## Author contributions

F.K. conducted literature review, created the model, collected data, performed numerical analysis, and wrote the manuscript. Z.Z. conducted literature review, validated the model, performed numerical analysis, and co-wrote the manuscript. M.R.D. designed the study, provided data and modeling tools, and co-wrote the manuscript.

## Competing interests

The authors declare no competing interests.
