## [Transparent Peer Review file · Nature Communications]

Coordinating Power Sector Climate Transitions Under Policy Uncertainty

Corresponding Author: Dr Michael Davidson

Version 0:

Reviewer comments:

Reviewer #1

(Remarks to the Author)

This paper performs modeling of the Western Interconnection to understand the impact of coordination on the cost of building out the power system in the west. The language is clear, and there are no obvious errors. My primary concerns with the manuscript are that 1) there is not enough detail provided to fully evaluate the scenario results and 2) the results might be claiming a little more than is merited, but it is hard to tell without more detail on the methods and scenario design. I recommend adding more detail and ensuring that the claims in the paper are aligned with the methods and scenario design employed.

Page 6, line 149: By “resources” do you mean “generation resources?” I consider transmission and storage as resources, but they are listed separately.

Page 6, line 128: You haven’t defined CES yet.

[I’m skipping to methods here, because I need to understand that more in order to interpret results]

Page 18, line 389: Do you include both offshore and land-based wind, or just land-based wind?

Page 19, lines 411-412: How many resource profiles do you have represented for wind and solar? You have a single cluster; is that one cluster per region, or one cluster for the entire model? I’m asking because the value of coordination depends on the resolution of the resource representation. If you include more heterogeneity of the resource, then there will be more value in building out transmission to the best resources, and hence more value in coordination.

Page 19, line 417-418: What happens if the net peak period shifts away from the peak load week, especially in the 100% CES scenarios? How do you know that you have maintained resource adequacy? Also, are you ensuring the resource adequacy is met locally? For example, the peak week might not be the most stressful period for Montana.

Relatedly, how do is RA enforced? You have the reserve margins in the SI, but are you enforcing them during all hours, just for the peak hours, or something else? Are you using probabilistic methods, or net load approximation methods, or something else? Are you using multiple weather years, or just one weather year? If one, then the value of coordination will likely be lower than if you look at many weather years.

Page 19, proposed transmission lines section: it would be nice if you had a pointer to the specific section of the SI here. I wanted a map or transmission line list to see what these lines were, and had to find them myself in the SI. That’s probably true of the other sections—including pointers to extra detail in the SI would be helpful to the reader.

Page 20, scenarios section: I don’t understand the scenarios, and in your study the scenario design is a key driver of the outcomes. It would be good to clarify the scenarios more, so that the results can be properly interpreted. For example, you say that the expanded EIM scenario changes to include all of WECC, but how is the EIM represented in the BAU? I thought that >70% of the WECC footprint was already in the EIM, so the delta from ~70% to 100% doesn’t seem very large, but

maybe you were representing no EIM in your BAU scenario?

As another example, you say you use hurdle rates in the BAU scenario, but I didn't see anywhere in the document where the hurdle rates are, and if they are applied across all regions for just across state boundaries. The SI also says they can be asymmetric, so knowing both the forward and reserve hurdle rates would be important.

And what is the difference between having resource adequacy be zonal instead of nodal? RA already needs to be enforced zonally (e.g., you can't only build capacity in WY and expect CA to be resource adequate unless you have the transmission to deliver the power to CA). Does it mean that you are restricting trading of RA capacity between zones? Or are you applying a hurdle rate to RA capacity trades? There is already RA trading among zones via ownership and bilateral agreements. For example, Idaho Power owns part of the Valmy coal plant, which is not in Idaho, but it counts toward their RA. The same goes for LADWP, which has significant assets outside of CA that are used to meet both RA and energy needs.

Said another way, if I wanted to try out your scenarios in my own energy model, I wouldn't be able to do it because the scenarios are not defined with sufficient information for me to be able to do that.

Table 1: I don't understand why operating reserves are at the zonal level in the BAU. Operating reserves are already shared over a larger region, which is why the requirement is smaller than if there actually no coordination already for operating reserves. If you wanted to truly capture a no-coordination element of operating reserves, you would need to include the operating reserve requirement because a single contingency event would represent a much larger share of the in-zone electricity demand.

Table 2: Arizona does not have a CES policy. See <https://emp.lbl.gov/publications/us-state-renewables-portfolio-clean>. And Colorado's CES only applies to Xcel energy, which cover just over half of the state's electricity. Finally, while most of these states have a nominal 100% CES, their effective limit is lower. For example, in California federal entities are not subject to the CES.

Page 21, lines 456-462: CES and RPS policies are almost always applied to sales, but you seem to be applying them to generation. E.g., losses from batteries, transmission, and distribution would not have to be clean because they are not subject to the policies. Maybe you are intending to simply model a carbon-free future, and if so, it would be best to characterize it that way rather than by using individual state policies, which as currently designed, will not lead to 100% clean energy.

Page 22, transmission planning section: Do you do anything to capture intrazonal transmission, from either accessing wind and solar sites, or for meeting higher load or more in-region generation?

Page 23, line 497: the use of "Laboratory" for the author of the ATB is rather strange. I don't think I've ever seen a national laboratory product cited that way.

Page 23: You note the NREL ATB as the source for your low cost sensitivity inputs, but I couldn't find anywhere what your default inputs were. Same for natural gas, coal prices, etc.

Methods section: I didn't see anywhere that explained how the system cost was calculated that you present in figure 2. For example, is the cost in nominal or real dollars? What kind of WACC or discount rate did you use for turning capital costs into annual costs? What economic lifetime are you using for that calculation? Where does the scope of that cost end?

Methods section: I couldn't find the load inputs. I looked through the SI as well, but didn't find them there. E.g., are you assuming a lot of electrification, or very little? Are load shapes different in 2050 than today?

Overarching comment on the methods and scenarios: I am concerned that the BAU scenario is not sufficiently stringent. Because the zones you have are fairly large, the model already has a lot of intra-region coordination because within the zone there are no transmission constraints or coordination challenges. And because there are way more balancing areas in the west that you have zones in your model, your aggregated approach inherently removes much of that friction. For example, the public utility districts in central Washington have to pay BPA hurdle rates to bring power from other parts of the state into their service territory, but that would not be captured in this approach, and thereby minimizes the impact of adding coordination on top of a zonally coordinated system. It seems like to truly answer the questions you posed, you would need to model each balancing areas as its own region. I imagine that would be out of scope, so I think you need to more carefully put this work in context of the assumptions you have made so that it can reflect the true findings.

From the SI: How much capacity is filtered out when you drop clusters less than 50 MW?

Does the model step through time, or does it solve for a single future year?

[Back to the results section now]

Page 8, lines 193-196: The 75% of in-state resources is different than the current law. For example, Washington allows resources within the Columbia River Basin to qualify. I think my hang-up with this is that you are talking about actual state RPS/CES policies, but you are applying them in generic ways, so it's hard to say what is representing the actual system versus what is an assumption you've made as a scenario design but deviates from the actual BAU.

For Figure 2 and the surrounding discussion, would you provide the total cost of operating the system so that we can better understand the relative magnitude of these numbers?

Page 9: you talk about firm capacity, but I don't think you ever defined it. Does 4-hour storage count as firm capacity? Do the existing CSP with storage plants count as firm capacity?

Figure 6: Can you make a companion chart for the SI that shows the relative amount of capacity change? I can't tell how big these numbers are from looking at the figure.

Page 15: The huge difference between your estimated benefits and EIMs calculated benefits seems really large. It seems like closing that gap would give more credibility to your study. At a minimum, I think it provide enough uncertainty that your claim of "We estimate the aggregate costs of fully coordinated power systems to be just a few percent lower than baseline (i.e., current) levels" is too weak to highlight this way in the abstract.

(Remarks on code availability)

The code is not available. The manuscript says it will be available upon publication.

Reviewer #2

(Remarks to the Author)

Thank you for the opportunity to review. The authors have developed a capacity expansion model of the western United States bulk electric system to study impacts of varying levels of regional market, transmission, and in-state generation requirements as region implements clean energy standards. From what I can tell the authors did an acceptable job summarizing the existing literature on this topic. The authors did a good job documenting the model and making datasets available.

Unfortunately, I found it difficult to follow the study's scenario design and presentation of results in the manuscript. It was not clear to me which aspects of the model (e.g. market scenario, transmission assumptions, in-state generation %) were changed across scenarios. This made it a challenge to interpret the meaning of authors' results and understand the comparative analysis produced from the model, and subsequently the authors' conclusions based on those modeling results. More specific points and questions along these lines are included below. If the authors were to revise this paper, I would recommend a large focus be on more clearly defining the scenario structure of the model and re-organizing the presentation of results accordingly.

- Please clarify how the model and/or application in this study differs from what was used in Davidson et al, 2024. Both studies appear to conclude that existing policies in the western US electricity sector raise costs by a "few percent" relative to a fully coordinated system.
- Section "Degrees of power system regionalization" may not need to be a stand-alone section, it could be condensed and combined with the introduction.
- Use of CES acronym beginnin gon page 6 line 158, I believe this means Clean Energy Standard, was this defined already? I may have missed it.
- Fig 1: Can you address balancing authority (BA) borders that aren't reflected in your model zones? BAs are a level of jurisdiction that coordinates transmission planning, generation dispatch, and resource adequacy coordination. Those dynamics may not be captured in your current zonal setup- for example, there are two balancing authorities within Colorado that have coordination frictions. The PacifiCorp BA coordinates these functions across most of Utah and Wyoming, and some of Idaho.

In general, I found it challenging to follow the basic structure and scenario design underlying the modeling and results. A few questions that arose for me:

- Regarding Fig 2: Why are the results for low-cost solar, battery, wind, and high-cost gas scenarios plotted as points, in addition to the cost ranges also plotted in this figure? Does a "low-cost solar" scenario hold the other resources' costs at "normal levels" and so on for the other technologies?
- What drives the cost ranges plotted in Fig 2? Perhaps it is some variation of transmission and/or generation coordination as you have defined it, but it's not clear which one or both are varied in the results in this chart. What do the different colors of the cost ranges represent (e.g. maroon, red, light blue)?
- In Fig 2, why does the right panel not include a 0% in-state RPS and CES scenario like the one on the left?
- Fig 3: what technologies for the non-emitting firm capacity are being built in the model?
- I am surprised by the authors' conclusion a 100% Regional CES scenario reduces exports across states, (p 12 line 239), is this relative to current state policy? I would expect higher levels of VRE to require more trade across the region to maintain balancing.
- What is "All" in Fig 5? Is that the scenario with existing clean energy policies?
 - o I don't think the results in this section accurately reflect policy uncertainty. The authors state that going from "All" to a 100% CES scenario means that "policy uncertainty disappears" (p 12 line 258). I disagree, there is significant uncertainty behind whether a 100% CES would be implemented in the Western United States.

(Remarks on code availability)

Reviewer #3

(Remarks to the Author)
see attached

(Remarks on code availability)

Version 1:

Reviewer comments:

Reviewer #1

(Remarks to the Author)

Thank you for the detailed responses to the reviewer comments. I feel like the manuscript is much improved from the original submission. A couple notes on the responses:

- With respect to the operating reserve sharing, I was referring to the shared requirement for the reserves (I recognize that wasn't clear). If each utility set its operating reserve requirement based on being an islanded system, the total requirements would be much higher. So there is already some intrinsic value in coordination/sharing that is part of the system. No need to address this in the manuscript, just trying to clarify the comment.
- I really like the addition of the sensitivity scenarios. I feel like that helps put your results in better context, and helps the reader understand the potential impact of the assumptions you have made along the way.
- I appreciate the pointers to the code and SI. I think I missed many of these the first time around simply because they weren't easy for me to find.

My biggest concern with the paper is still that the baseline scenario that you are measuring against is too coordinated relative to real life, largely because of the large-ish region sizes you are using. There is still plenty of "within-region" friction that is present today that cannot be captured by this model at its current spatial resolution. In your discussion section on page 19 I recommend making your discussion of the limitation of in-state friction a little more clear, perhaps by noting that all western states have more than one utility, so in-state frictions between those utilities are not captured in the scenarios modeled.

(Remarks on code availability)

Reviewer #2

(Remarks to the Author)

I reviewed the authors' revisions in response to my report and am OK with how they have addressed my peer review

(Remarks on code availability)

Response to Reviewers:

“Coordinating Power Sector Climate Transitions Under Policy Uncertainty”

Fikri Kucuksayacigil, Zhenhua Zhang and Michael R. Davidson

October 24, 2024

Legend:

Black text – Reviewer comments

Red text – Response to reviewers

Blue text – New text added to paper

REVIEWER COMMENTS

Reviewer #1 (Remarks to the Author):

This paper performs modeling of the Western Interconnection to understand the impact of coordination on the cost of building out the power system in the west. The language is clear, and there are no obvious errors. My primary concerns with the manuscript are that 1) there is not enough detail provided to fully evaluate the scenario results and 2) the results might be claiming a little more than is merited, but it is hard to tell without more detail on the methods and scenario design. I recommend adding more detail and ensuring that the claims in the paper are aligned with the methods and scenario design employed.

Thank you for your comments. We added more explanations throughout the paper to clarify scenario design and improve reproducibility.

Page 6, line 149: By “resources” do you mean “generation resources?” I consider transmission and storage as resources, but they are listed separately.

We mean “generation resources”. We do not consider transmission and storage among resources. We revised that sentence.

Page 6, line 128: You haven’t defined CES yet.

Thank you for pointing this out. We have now defined CES where it appears first.

[I'm skipping to methods here, because I need to understand that more in order to interpret results]

Page 18, line 389: Do you include both offshore and land-based wind, or just land-based wind?

We include both in our model. We revised that sentence to emphasize each separately.

Page 19, lines 411-412: How many resource profiles do you have represented for wind and solar? You have a single cluster; is that one cluster per region, or one cluster for the entire model? I'm asking because the value of coordination depends on the resolution of the resource representation. If you include more heterogeneity of the resource, then there will be more value in building out transmission to the best resources, and hence more value in coordination.

Thank you for raising this point. We have 105 existing generator clusters in our data set across WECC. Each modeling zone has one cluster to represent a distinct resource type (natural gas, biopower, solar, wind, etc.). Hence, existing solar units are aggregated into one cluster for each modeling zone (similarly for wind). For new-build resources, we have more granularity in terms of resource profiles for wind and solar (375 clustered resources in total). Each modeling zone has multiple resource profiles for solar and wind. For instance, Southern California has 36 clusters of utility PV and 11 clusters of land-based wind.

For the revised manuscript, we added the following paragraph in the Methods section: "Out of 480 clusters, existing generators represent 105 clusters. Each modeling zone has one cluster to represent a distinct resource type (natural gas, biopower, solar, wind, etc.). Existing solar and wind resources are represented with a single cluster in each zone. New-build resources represent 375 clusters. New-build solar and new-build wind resources have multiple clusters, hence multiple resource profiles can be chosen by the model."

Page 19, line 417-418: What happens if the net peak period shifts away from the peak load week, especially in the 100% CES scenarios? How do you know that you have maintained resource adequacy? Also, are you ensuring the resource adequacy is met locally? For example, the peak week might not be the most stressful period for Montana.

We utilize PowerGenome to sample representative weeks from a whole year as input load data, described further under "Time sampling" in Methods. PowerGenome finds peak load time points and includes the corresponding week in the data, without consideration of renewable generation output. Hence, we do not use net peak load in our model.

We model resource adequacy in the following way: Our resource adequacy scenarios make different assumptions on the zonal aggregation. For BAU and Expanded EIM scenarios, we compute peak load plus a planning reserve margin to find the required capacity for each modeling zone. The model's resource adequacy constraints ensure that available resources, adjusted by capacity factor, are enough to meet the load by a margin. We consider fixed import and export in these equations (import and export are increased in Expanded EIM scenarios, see Expressions (93) and (94) in supplementary information). Hence, if the system is constrained in peak net load periods outside of the sample period, our model may not capture these constraints. This is generally unavoidable with time sampling where renewable energy deployment is endogenous. In the Regional Market scenario, the inequality is even simpler. We find the regional peak load magnitude and add a margin to find the load to meet. The model optimizes the resource capacities in the region to reach the load. We use the same capacity factors to adjust the resource capacities (refer to Expression (95) in supplementary information).

We revised the Methods section to clarify the scenario design. We also put the following explanation in the Methods section: "Peak weeks are computed separately by zone (in the zonal resource adequacy scenario) or a single region-wide peak week (in the regional resource adequacy scenario)."

Relatedly, how is RA enforced? You have the reserve margins in the SI, but are you enforcing them during all hours, just for the peak hours, or something else? Are you using probabilistic methods, or net load approximation methods, or something else? Are you using multiple weather years, or just one weather year? If one, then the value of coordination will likely be lower than if you look at many weather years.

(i) We enforce RA constraints only for peak load hours (ii) We do not do probabilistic modeling or net load approximation modeling. It is possible to integrate outage distribution or uncertainty quantification into the current model to estimate metrics of Expected Unserved Energy or Loss-of-Load Expectation (Stephen, 2021), but it requires a separate simulation module whose results are fed into the current linear optimization model. While it is possible to do this in the future, we maintain a self-contained linear optimization model at the moment to enforce RA constraints. (iii) We use one weather year (2012) available in PowerGenome. In the Methods section, we clarified planning reserves and peak week above. We clarified weather year in Methods:

"Demand and renewable energy profiles are taken from 2012."

We also added an explanation in the Discussion section as follows: "Our modeling results rely on sampled weeks (including peak week) from a single year. If the model

were run for multiple years, this may lead to higher cost savings for the regional market relative to the BAU.”

Page 19, proposed transmission lines section: it would be nice if you had a pointer to the specific section of the SI here. I wanted a map or transmission line list to see what these lines were, and had to find them myself in the SI. That’s probably true of the other sections—including pointers to extra detail in the SI would be helpful to the reader.

Thank you for your suggestions. We added these pointers wherever related in the manuscript.

Page 20, scenarios section: I don’t understand the scenarios, and in your study the scenario design is a key driver of the outcomes. It would be good to clarify the scenarios more, so that the results can be properly interpreted. For example, you say that the expanded EIM scenario changes to include all of WECC, but how is the EIM represented in the BAU? I thought that >70% of the WECC footprint was already in the EIM, so the delta from ~70% to 100% doesn’t seem very large, but maybe you were representing no EIM in your BAU scenario?

Thank you for the comment. We added substantially more explanation throughout the paper to further clarify scenario design.

Briefly, the BAU scenario assumes that there is low coordination outside of CAISO. Electricity is still traded between states, but with bilateral contracts which leads to jurisdictional frictions modeled with hurdle rates. Operating reserves and resource adequacy are met at the zonal level. The Expanded EIM scenario assumes that the current WEIM expands to the entire WECC. Energy trade becomes frictionless, operating reserves are met at the regional level, and resource adequacy is met at the zonal level with increased import/export capabilities. Hence, we do not explicitly consider the existing footprint of WEIM in our model. Scenario descriptions are summarized in Table 2 and Table 3.

As another example, you say you use hurdle rates in the BAU scenario, but I didn’t see anywhere in the document where the hurdle rates are, and if they are applied across all regions or just across state boundaries. The SI also says they can be asymmetric, so knowing both the forward and reserve hurdle rates would be important.

We apologize for the difficulty in ascertaining these assumptions. We listed hurdle rates in Supplementary Table 17, and we now point to this table in the Methods section of the revised manuscript. They can be asymmetric as they depend on the exporting region. For instance, while power flowing from Arizona to Southern California is subject to \$7.35/MWh, power flowing from Southern California to Arizona incurs a \$10.39/MWh hurdle rate. As we explained in the supplementary information, we do not enforce any

hurdle rate for the power flowing between California modeling zones. Hurdle rates are 0 in the Expanded EIM and Regional Market scenarios, reflecting energy market coordination. We put another paragraph in the Methods section: “We apply hurdle rate costs across zones based on the exporting region (see Section 5.14 of Supplementary Information). There is no hurdle rate for trade between California zones.”

Due to the difficulty in assessing hurdle rates, we now include sensitivity analysis on hurdle rates (described below).

And what is the difference between having resource adequacy be zonal instead of nodal? RA already needs to be enforced zonally (e.g., you can’t only build capacity in WY and expect CA to be resource adequate unless you have the transmission to deliver the power to CA). Does it mean that you are restricting trading of RA capacity between zones? Or are you applying a hurdle rate to RA capacity trades? There is already RA trading among zones via ownership and bilateral agreements. For example, Idaho Power owns part of the Valmy coal plant, which is not in Idaho, but it counts toward their RA. The same goes for LADWP, which has significant assets outside of CA that are used to meet both RA and energy needs.

Resource adequacy constraints can be enforced at the zonal level, which in our model corresponds to 12 regions roughly at the state level. Indeed, resource adequacy is complicated and a fuller consideration would involve examining the individual decisions of balancing authorities. Prior to considering zonal resource adequacy, we account for regular export/import relationships through long-term contracting (as reported by annual electricity transfer data from the Western Electricity Coordinating Council and summarized in Table 11 and Table 13 of our supplementary information), which effectively counts some out-of-state resources toward meeting zonal RA as you describe.

Said another way, if I wanted to try out your scenarios in my own energy model, I wouldn’t be able to do it because the scenarios are not defined with sufficient information for me to be able to do that.

Thank you for your suggestions. We further elaborated the scenario description throughout the revised manuscript and summarized the description in Table 2 and Table 3. The specific RA constraints and contributions are elaborated in Expressions (93), (94), and (95) of our supplementary information and in `planning_reserve.csv` and `Load_data.csv` of our data repository (margin and fixed import/export information from the first file and load information from the second file).

Table 1: I don’t understand why operating reserves are at the zonal level in the BAU. Operating reserves are already shared over a larger region, which is why the requirement is smaller than if there actually is no coordination already for operating reserves. If you wanted to truly capture a

no-coordination element of operating reserves, you would need to include the operating reserve requirement because a single contingency event would represent a much larger share of the in-zone electricity demand.

According to our understanding (and elaborated in Energy Strategies, 2021a, 2021b), reserves are not currently shared across the WECC, and the EIM only involves short-term energy balancing. Perhaps this is in reference to the Reserve Sharing Program established in the Northwest Power Pool. It initially started prior to 2024, and it spanned some parts of the Pacific Northwest and Canada. Later in 2024, some other balancing authorities from WECC joined this group (DiFabio, 2024) (CAISO, LADWP and CFE – Mexico's Comisión Federal de Electricidad – are still not in this group). The group shares some contingency reserves.

Such a large coordination in sharing contingency reserves was not existing at the time of writing this paper (and available coordination was very limited when we started writing the paper) (Catchpole, 2024). Also, given that CAISO, LADWP and CFE are still not part of this program, the overall level of reserve coordination is still limited.

We run an additional sensitivity analysis on several factors and report results in the manuscript: in-state CES/RPS fractions, operating reserve requirements, and hurdle rates. This sensitivity analysis reveals that our results do not change significantly when we increase the operating reserve requirements.

Summary Table: Sensitivity Analysis

Sensitivity factors	In-state CES/RPS ratios	Operating reserve requirements	Hurdle rates
State Policy Scenario			
Incomplete Coordination	0%, 25%, 50%, 75% (baseline), 87.5%, 100%	3%+5% (baseline), 3%+10%, 5%+10%	CEC (baseline), CEC low, CPUC high
BAU	0%, 25%, 50%, 75% (baseline), 87.5%, 100%	3%+5% (baseline), 3%+10%, 5%+10%	CEC (baseline), CEC low, CPUC high
Expanded EIM	0%, 25%, 50%, 75% (baseline), 87.5%, 100%	3%+5% (baseline), 3%+10%, 5%+10%	N/A

Regional Market	0%, 25%, 50%, 75% (baseline), 87.5%, 100%	3%+5% (baseline), 3%+10%, 5%+10%	N/A
Full Coordination	N/A	3%+5% (baseline), 3%+10%, 5%+10%	N/A
Regional 100% CES Scenario			
BAU	N/A	3%+5% (baseline), 3%+10%, 5%+10%	CEC (baseline), CEC low, CPUC high
Expanded EIM	N/A	3%+5% (baseline), 3%+10%, 5%+10%	N/A
Regional Market	N/A	3%+5% (baseline), 3%+10%, 5%+10%	N/A
Count	24 cases	24 cases	9 cases

Factor 1: In-state CES/RPS ratios

In the baseline, we assume that 75% of the clean generation should be delivered from in-state clean resources to meet CES/RPS targets under the State Policy scenario (Perez et al., 2016). In other words, the remaining 25% of the clean generation can be supplied using out-of-state Renewable Energy Credits (RECs). In addition, we examine how varying in-state CES/RPS ratios will impact our findings—0%, 25%, 50%, 75% (baseline), 87.5%, and 100%, similar to the approach in Perez et al. (2016). If in-state CES/RPS ratios are 100%, this corresponds to the least flexible case in terms of REC trading.

Factor 2: Operating reserve requirements

In the baseline, we adopt the “3+5” heuristic, meaning that operating reserves (both up and down) have to be no less than 3% of the load and 5% of the renewable output at each hour, as described in the Western Wind and Solar Integration Study and other modeling studies (National Renewable Energy Laboratory, 2017, Johnston et al., 2019). The exact operating reserve requirements in practice vary by product and market. Sergi and Cole (2021) calculate the reserve requirement levels based on methods from the Western Electricity Coordinating Council Transmission Expansion Planning Policy Committee (WECC TEPPC) (Sergi and Cole, 2021, Lew et al., 2013). In the report,

spinning reserves have to be no less than 3% of the load, and ramping reserves have to be no less than 10% of wind generation and 4% of solar generation.

Due to the uncertain and variable nature of wind and solar resources, high VRE penetrations lead to increases in reserves that are necessary for the power systems (Lew et al., 2013). Hence, we examine higher levels of operating reserve requirements under all the regionalization and policy scenarios: (1) 3% of the load and 5% of the renewable output (baseline), (2) 3% of the load and 10% of the renewable output, and (3) 5% of the load and 10% of the renewable output.

Factor 3: Hurdle rates

We use “hurdle rates” as a proxy for the inter-jurisdictional transaction barriers. The hurdle rate costs effectively increase the trading costs between two non-coordinating zones. The baseline cost assumptions are obtained from the CEC SB100 Joint Agency Report (California Energy Commission, 2020). Furthermore, we consider two additional sets of hurdle rates: (1) alternative rates, and (2) low rates. We obtain the alternative rates based on the CPUC System Reliability Modeling Datasets (California Public Utilities Commission, 2023), and the low rates by discounting the baseline rates by 50%.

Table for hurdle rate sensitivity (\$/MWh)

Zones	Baseline (CEC)	Alternative rates (CPUC)	Low rates (CEC low)
Arizona	7.35	4.6215	3.675
Colorado	4.91	5.8266	2.455
Idaho	4.91	3.1239	2.455
Montana	4.91	5.5458	2.455
New Mexico	7.35	4.8672	3.675
Nevada	7.35	8.1432	3.675
North California	10.39	12.6945	5.195
Pacific Northwest	4.91	3.6036	2.455
San Diego	10.39	12.6945	5.195

South California	10.39	12.6945	5.195
Utah	4.91	3.6036	2.455
Wyoming	4.91	5.8266	2.455

We added the following paragraph in the Methods section of the revised manuscript:

“We perform sensitivities to evaluate the robustness of results around key regionalization modeling parameters, namely, in-state fractions of RPS/CES goals, operating reserve requirements, and hurdle rates. We run all scenarios given in Figure 2 with different values for these parameters. A detailed explanation of the sensitivity analysis design is given in Supplementary Information.”

We added the following paragraph in the main text just before the Discussion section:

“We conduct additional sensitivity analysis by running all scenarios with different in-state fractions of RPS/CES goals (default/baseline values are 0% for Full Coordination and 75% for all other market coordination scenarios), operating reserve requirements (default is “3+5” heuristic), and hurdle rates (see Methods and Supplementary Information for more details). Decreasing in-state CES/RPS requirements (i.e., increasing REC flexibility) from 100% to 0% leads to approximately \$200M in cost savings -- about 0.7% of the system cost in the BAU scenario -- across all scenarios due to less wind and solar deployment. Higher operating reserve requirements lead to increases in system costs of less than \$90M due to the additional storage deployment. Varying hurdle rate costs can lead to system cost savings of up to \$180M -- about 0.5% of the system cost in the BAU scenario -- driven by the additional solar deployment in low-rate zones and less wind deployment in high-rate zones. These indicate that our main results are robust to the selected modeling parameters because the change in the system cost is only about less than 1% of the system cost in the BAU scenario. As indicated before, the aggregate system costs of fully coordinated power systems remain just a few percent lower than baseline levels, with large region-wide differences.”

Table 2: Arizona does not have a CES policy. See

<https://emp.lbl.gov/publications/us-state-renewables-portfolio-clean>. And Colorado’s CES only applies to Xcel energy, which covers just over half of the state’s electricity. Finally, while most of these states have a nominal 100% CES, their effective limit is lower. For example, in California federal entities are not subject to the CES.

Thank you so much for raising these issues. We were aware of them when we set these targets for Arizona and Colorado, though our description of the state policy scenarios was inadequate. To clarify, we now refer to this scenario as “Continued State Policy”

and assign 100% CES targets to states being on track to do so. In particular, we decided to implement 100% CES targets for three additional states because of the following reasons:

Even though the Arizona Corporation Commission rejected a 100% CES by 2050 in 2021, there was much discussion about it. At the same time, there have been pledges from the three main utility companies: Arizona's largest utility, APS (~1.3m households), announced 100% CES by 2050, Tucson Electric Power (~0.4m households) plans to reduce carbon emissions by 80% by 2035, and have a 70% RPS by 2035, and Salt River Project (~1.1m customers) plans to reduce carbon by 65% in 2035 and 90% by 2050. Hence, we think a continuation of these pledges will work *de facto* similar to a 100% CES which would be codified by the state.

It is correct that Colorado SB19-263 requires 100% CES by 2050 for utilities serving 500k or more customers, which then only applies to Xcel (which takes up ~60% of the state's load) (Clean Energy States Alliance, 2024). However, Colorado also has an SB16 that requires the whole *economy* to reduce GHG emissions by 100% by 2050. Hence, we think it is reasonable to assume that the power sector has a *de facto* 100% CES target (National Caucus of Environmental Legislators, 2023).

Regarding California, the net-zero target set for 2045 applies to all activities and sectors within the state (e.g., electricity production, transportation, buildings, agriculture, etc.). Energy consumption by federal entities in California accounted for 2% of total energy consumption in the state in 2000 (National Economic Council, 2000). Unfortunately, we do not have recent data for this statistic. In terms of power generation, the federally-managed Western Area Power Administration (WAPA) operates some power plants in California—though, these are zero-emitting hydropower resources; hence, they automatically meet CES requirements.

We added a paragraph in the Methods section of the revised manuscript as follows: “The Continued State Policy scenario is more optimistic than current formal targets in some cases. If major utilities in the state have adopted 100% CES or similar targets, or if the state’s other climate goals create a *de facto* CES, then we adopt 100% CES for that state. These are elaborated in Section 7 of the Supplementary Information.”

Page 21, lines 456-462: CES and RPS policies are almost always applied to sales, but you seem to be applying them to generation. E.g., losses from batteries, transmission, and distribution would not have to be clean because they are not subject to the policies. Maybe you are intending to simply model a carbon-free future, and if so, it would be best to characterize it that way rather than by using individual state policies, which as currently designed, will not lead to 100% clean energy.

It is correct that we apply CES and RPS policies on generation.

In the literature, there are examples of CES and RPS policies being applied to either generation or sales. Under the condition that transmission losses are ignored and unserved energy is penalized with a sufficiently high cost, these policies can be applied to generation instead of utility sales (Munoz et al., 2013). We in fact tested various forms of policy constraints in our model. We prefer a generation-based approach because (i) load-based constraints are more computationally challenging, (ii) we assume transmission losses are negligible, and (iii) the penalty for unserved energy is sufficiently high (\$603/MWh up to 7.5% of demand and \$9000/MWh up to 100% of demand).

Page 22, transmission planning section: Do you do anything to capture intrazonal transmission, from either accessing wind and solar sites, or for meeting higher load or more in-region generation?

We only focus on expanding inter-zonal transmission lines. Even though we ignore intra-zonal transmission line expansion, PowerGenome provides the total cost of transmission expansion as a sum of capital cost of expansion and the cost of connecting generators to the main grid. Hence, we capture intra-zonal transmission expansion at a system-wide cost. We put another paragraph in the Methods section as follows: “We only focus on inter-zonal transmission line expansion in this study. While we omit explicit intra-zonal line expansion, we integrate their costs (e.g., connecting generators to the main grid) in inter-zonal line expansion.”

Page 23, line 497: the use of “Laboratory” for the author of the ATB is rather strange. I don’t think I’ve ever seen a national laboratory product cited that way.

Thank you for spotting this error. We used Latex to compile the document and have corrected the bibliography file.

Page 23: You note the NREL ATB as the source for your low cost sensitivity inputs, but I couldn’t find anywhere what your default inputs were. Same for natural gas, coal prices, etc.

Default cost numbers can be found in the shared data repository folder (WECC-data). It is located in the *Inv_Cost_per_MWyr* column of *WECC-data/powergenome_output/reference_cost/Generators_data.csv*. Cost numbers used for sensitivity analysis can be found in the same files located in the *low_cost_wind*, *low_cost_solar*, *low_cost_battery*, and *high_cost_gas* folders. We add an explanation in

the Methods section as follows: “For cost data, see generator data files in our repo corresponding to reference cost cases and sensitivity analysis.”

Methods section: I didn’t see anywhere that explained how the system cost was calculated that you present in figure 2. For example, is the cost in nominal or real dollars? What kind of WACC or discount rate did you use for turning capital costs into annual costs? What economic lifetime are you using for that calculation? Where does the scope of that cost end?

The objective function of the model represents the system cost, which is shown in Expressions (102a) - (102g) of supplementary information. We added a paragraph in the Methods section as follows: “Objective function: Objective function minimizes system-wide annual cost. It includes annualized investment and fixed costs of resources, storage units, and transmission paths, variable cost of generators, cost of non-served demand, startup costs of thermal plants, and hurdle rate costs in the Incomplete Coordination and BAU scenario. See Section 3 of Supplementary Information for more details.”

PowerGenome provides the cost numbers used in our model. We use annualized investment cost (per MW and MWh) and annualized fixed cost (per MW and MWh) for resources and storages, annualized expansion and fixed costs for transmission paths, variable costs for resources (as a function of start-up fuel cost of thermal units and fuel costs of all units), start-up cost of thermal units, and cost of non-served energy (penalized with \$603/MWh up to 7.5% of demand, and \$9000/MWh up to 100% of demand). Please see *Generators_data.csv*, *Network.csv*, and *demand_segments.csv* files for all of our cost numbers. PowerGenome extracts cost numbers from NREL ATB. According to raw data provided by NREL ATB, capital expenditure costs are in 2018 dollars (so, cost numbers are nominal values). WACC numbers vary across resources between 5% and 7%. We assume a 20-year economic lifetime for resources. We added the following clarification in the Method section: “...(cost values are in 2018 dollars for both reference and sensitivity)..”

Methods section: I couldn’t find the load inputs. I looked through the SI as well, but didn’t find them there. E.g., are you assuming a lot of electrification, or very little? Are load shapes different in 2050 than today?

Load data are available in the data supplement, specifically in *Load_MW_** columns of *WECC-data/powergenome_output/reference_cost/Load_data.csv*. We do not assume electrification because we want our model to represent a baseline (e.g., high electrification would be an additional scenario). The load shape in 2050 is not different from today’s load shape. We added a new paragraph in the Methods section as follows:

“Hourly load data over the sampled six weeks is taken from PowerGenome and includes baseline electrification levels.”

Overarching comment on the methods and scenarios: I am concerned that the BAU scenario is not sufficiently stringent. Because the zones you have are fairly large, the model already has a lot of intra-region coordination because within the zone there are no transmission constraints or coordination challenges. And because there are way more balancing areas in the west that you have zones in your model, your aggregated approach inherently removes much of that friction. For example, the public utility districts in central Washington have to pay BPA hurdle rates to bring power from other parts of the state into their service territory, but that would not be captured in this approach, and thereby minimizes the impact of adding coordination on top of a zonally coordinated system. It seems like to truly answer the questions you posed, you would need to model each balancing area as its own region. I imagine that would be out of scope, so I think you need to more carefully put this work in context of the assumptions you have made so that it can reflect the true findings.

Thank you for your comment. The spatial and temporal granularity of the power system optimization models is always a concern and relates to the computational tractability of the model and availability of the data. Unfortunately, we do not have the same level of granularity for all data types. Additionally, modeling state policy becomes more complicated and bias-prone when we attempt to model zones at higher granularity, because RPS/CES constraints are not at the BA level (elaborated below). We have opted for an open-source package, PowerGenome, to obtain input data for the model which also aligns approximately with policy boundaries.

We run an additional sensitivity analysis for hurdle rate costs for the revised manuscript. You can review our response above, supplementary information, and additional text in the manuscript regarding our findings. It shows that when we increase hurdle rate costs to make the BAU scenario more stringent, total system cost increases by less than 1%.

We plan to extend this work by integrating additional market coordination scenarios, including ones with in-state frictions. In the revised manuscript, we add the following explanation in the Discussion section where we address the limitation of our work and future extensions: “We utilize aggregated modeling zones which ignore in-state frictions. Higher spatial granularity could be integrated to estimate the value of coordination more accurately, though with important computational trade-offs.”

From the SI: How much capacity is filtered out when you drop clusters less than 50 MW?

We use a 50 MW threshold to drop resources in two places. First, we drop existing biomass resources whose size is less than 50 MW. This operation removes biomass resources in six modeling zones whose total capacity is 53 MW. Second, we drop

candidate (new build) onshore and offshore wind as well as utility PV resource clusters whose size is less than 50 MW. This operation removes 18 clusters of these candidate resources over four modeling zones, with a total capacity of 509 MW.

Does the model step through time, or does it solve for a single future year?

The model solves for a single future year (2050).

[Back to the results section now]

Page 8, lines 193-196: The 75% of in-state resources is different from the current law. For example, Washington allows resources within the Columbia River Basin to qualify. I think my hang-up with this is that you are talking about actual state RPS/CES policies, but you are applying them in generic ways, so it's hard to say what is representing the actual system versus what is an assumption you've made as a scenario design but deviates from the actual BAU.

We appreciate the attention to specific policy details. In a capacity expansion model with zonal granularity, it would not be feasible to include separate regulatory requirements at the sub-zonal level. We capture as much detail as we can in terms of the eligible resources and percentage requirements as well as assumptions of policy continuation (explained above as a response to the comment about Table 2), though there are necessarily some simplifications. As we are modeling the year 2050, we also feel it may inject too much over-precision to assume that specific regulatory requirements remain unchanged over the intervening two and half decades. Therefore, we aim to have the broad policies correct and to generate scenarios that allow for more interpretable conclusions. For example, we could set different in-state requirements by zone, though we chose a single parameter setting because there is no readily available source with the specific numbers, and to avoid further justifications about the persistence of these ratios going forward (see Munoz et al., 2017 and Perez et al., 2016 for similar assumptions in the literature).

We address some of these concerns by running sensitivity analysis for in-state fractions of RPS and CES goals for the revised manuscript. We report our results above. Our results show small cost differences when we change this parameter from 0% and 100% is only within 0.7% of the total system cost in the BAU scenario, indicating that generation coordination has a small impact on the system-wide cost. When we increase in-state fraction from 0% to 100% (e.g., decreasing the flexibility of REC trade), (i) the states' investment in wind resources show a dramatic increase due to its high potential; (ii) low-ambition states increase the investment in non-CCS gas resources, and (iii) region-wide costs of generation investments and the non-served energy increase. However, when we increase in-state fraction from 0% to 100%, no substantial change is

observed in the distribution of resources over the region if transmission coordination is limited.

For Figure 2 and the surrounding discussion, would you provide the total cost of operating the system so that we can better understand the relative magnitude of these numbers?

Thank you for your suggestion. The cost of State Policy + BAU (the second bar in the left panel) is \$36.73B/year. The cost of Regional 100% CES + BAU (the first bar in the right panel) is \$41.98B/year. We added these numbers in the first paragraph of the Results section: “...(for reference, cost of State Policy + BAU, the second bar in the left panel of Figure 2, is \$36.73B/year and cost of Regional 100% CES + BAU, the first bar in the right panel, is \$41.98B/year)...”

Page 9: you talk about firm capacity, but I don't think you ever defined it. Does 4-hour storage count as firm capacity? Do the existing CSP with storage plants count as firm capacity?

Firm capacity includes biopower, non-CCS coal, CCS coal, geothermal, non-CCS gas, CCS gas, hydro, nuclear, and other small-sized plants such as municipal waste. We do not include battery and CSP as firm capacities. We added the following text in the first paragraph of the Results section as follows: “...(we define firm resources as biopower, non-CCS coal, CCS coal, geothermal, non-CCS gas, CCS gas, hydropower, nuclear, and other small-sized plants such as municipal waste)...”

Figure 6: Can you make a companion chart for the SI that shows the relative amount of capacity change? I can't tell how big these numbers are from looking at the figure.

We added Supplementary Figure 20 in the supplementary information. We also added a sentence around Figure 6 in the manuscript which refers to Supplemental Figure 20 as follows: “...(see Supplementary Figure 20 as complementary to Figure 6, where we show reference cost case capacity and relative change on top of those capacity values).”

Page 15: The huge difference between your estimated benefits and EIMs calculated benefits seems really large. It seems like closing that gap would give more credibility to your study. At a minimum, I think it provides enough uncertainty that your claim of “We estimate the aggregate costs of fully coordinated power systems to be just a few percent lower than baseline (i.e., current) levels” is too weak to highlight this way in the abstract.

As we stated in the manuscript, the real benefit of EIM is related to extreme conditions (highlighted in the “Discussion and conclusion” as strong hydro conditions, high gas prices, and increased demand), which we did not simulate in our study. Also, the Expanded EIM scenario does not reflect the current EIM. It reflects an EIM with

increased import and export capabilities (many more gainings would also be included here in addition to increased import and export, but it was out of scope). Hence, we believe that comparing the actual benefit of the current EIM with the estimated benefit of the Expanded EIM does not fully make sense. We gave these numbers in the manuscript just to give a sense of how big our estimation numbers are relative to the real cost savings of the current EIM. We revised the sentence in the Abstract highlighting this point as follows: “We estimate the aggregate costs of fully coordinated power systems (ignoring extreme conditions) to be just a few percent lower than baseline (i.e., current) levels, noting that state-level differences can be an order of magnitude larger.”

Reviewer #1 (Remarks on code availability):

The code is not available. The manuscript says it will be available upon publication.

We apologize for the difficulty. We uploaded data and code files as zip folders in the submission and will post a public repo upon acceptance. We have uploaded the same folders along with the revised manuscripts in this submission.

Reviewer #2 (Remarks to the Author):

Thank you for the opportunity to review. The authors have developed a capacity expansion model of the western United States bulk electric system to study impacts of varying levels of regional market, transmission, and in-state generation requirements as the region implements clean energy standards. From what I can tell the authors did an acceptable job summarizing the existing literature on this topic. The authors did a good job documenting the model and making datasets available.

Thank you for your comments.

Unfortunately, I found it difficult to follow the study’s scenario design and presentation of results in the manuscript. It was not clear to me which aspects of the model (e.g. market scenario, transmission assumptions, in-state generation %) were changed across scenarios. This made it a challenge to interpret the meaning of authors’ results and understand the comparative analysis produced from the model, and subsequently the authors’ conclusions based on those modeling results.

Thank you for your feedback. We revised the manuscript to clarify the scenario design (see response to Reviewer 1 above).

More specific points and questions along these lines are included below. If the authors were to revise this paper, I would recommend a large focus be on more clearly defining the scenario structure of the model and re-organizing the presentation of results accordingly.

We carefully responded to your questions below. We hope that the scenario design and its link to results and policy implications are now much clearer to you in the revised version.

- Please clarify how the model and/or application in this study differs from what was used in Davidson et al, 2024. Both studies appear to conclude that existing policies in the western US electricity sector raise costs by a “few percent” relative to a fully coordinated system.

Davidson et al. (2024) analyze sustainability systems from the perspective of how they behave under the rules and laws set by regulatory bodies. It takes an example of climate mitigation systems and examines three classes of models – integrated assessment model, agent-based model, and engineering-economic optimization (EEO) model. For EEO, a mixed-integer linear optimization model is built; optimizing resource, transmission, and storage capacities in eleven states of the western U.S. as well as optimizing the hourly operation of the power system through one year. By assuming a Regional 100% CES scenario, the power system with institutional heterogeneity is compared to the power system with no institutional friction. In the current paper, the setting of Davidson et al. (2024) is expanded substantially. First, we consider different levels of state policy in 2050, which is crucial since many states do not have a target or likely target of 100% CES in 2050. Second, we add new market scenarios in order to assess the individual impacts of different coordination features. Third, we add generation and transmission coordination in order to provide a more complete picture beyond standard market operations. Similarly, we make conclusions about coordination at a broader level. Lastly, we conduct extensive sensitivity analysis to ascertain robustness of findings.

We agree that one of the common results from both papers is to see a few percent increase in system-wide cost from a coordinated system to an uncoordinated system. The novelty of the current paper’s findings relates to the different power system expansion behaviors when states follow individual state policies instead of a regional policy, regional benefits gained from different market formations including the coordination in transmission and generation planning, and states’ robust investment decisions. In the current paper, we find and discuss how storage and transmission are complementary, how investment locations for renewable resources change under different market and policy scenarios, and which robust decisions for solar, wind, and storages states should make to hedge against the market and policy scenarios.

- Section “Degrees of power system regionalization” may not need to be a stand-alone section, it could be condensed and combined with the introduction.

We combined that section with the Introduction in the revised manuscript.

- Use of CES acronym beginning on page 6 line 158, I believe this means Clean Energy Standard, was this defined already? I may have missed it.

We defined it on Page 7 in the revised version.

- Fig 1: Can you address balancing authority (BA) borders that aren’t reflected in your model zones? BAs are a level of jurisdiction that coordinates transmission planning, generation dispatch, and resource adequacy coordination. Those dynamics may not be captured in your current zonal setup- for example, there are two balancing authorities within Colorado that have coordination frictions. The PacifiCorp BA coordinates these functions across most of Utah and Wyoming, and some of Idaho.

It is correct that our zonal model does not consider boundaries and corresponding frictions between balancing authority within the same modeling zone. Including higher spatial granularity is possible, but we are limited by data availability (e.g., PowerGenome and the incompleteness of utility-level data) and computational tractability. Furthermore, setting of zones roughly on the state level better corresponds to policy, thereby avoiding some more difficult modeling decisions about how to achieve state-wide regulatory requirements at the sub-state utility or BA level, especially for states that do not have 100% CES or RPS requirements (elaborated above in response to Reviewer 1). As for Figure 1, we put an explanation within the last paragraph prior to the Results section as follows: “...this model does not consider boundaries and corresponding frictions between balancing authorities within the same modeling zone...”

In general, I found it challenging to follow the basic structure and scenario design underlying the modeling and results. A few questions that arose for me:

- Regarding Fig 2: Why are the results for low-cost solar, battery, wind, and high-cost gas scenarios plotted as points, in addition to the cost ranges also plotted in this figure? Does a “low-cost solar” scenario hold the other resources’ costs at “normal levels” and so on for the other technologies?

This is correct: low-cost solar, battery, wind, and high-cost gas are sensitivity analyses. Our scenarios (main bars in Figure 2) refer to different configurations of clean energy targets and regionalization. For a given scenario, we then change the costs of solar, wind, battery, and natural gas for the shown sensitivity analyses. The dots in Figure 2 refer to sensitivity cost results and the main bars refer to scenario cost results. We

added a short explanation within Figure 2 text as follows: “Bars represent scenarios of regionalization and clean energy policy. Dots represent sensitivity analyses keeping the model structure the same.”

- What drives the cost ranges plotted in Fig 2? Perhaps it is some variation of transmission and/or generation coordination as you have defined it, but it’s not clear which one or both are varied in the results in this chart. What do the different colors of the cost ranges represent (e.g. maroon, red, light blue)?

As we show in the x-axis, costs pertain to scenarios. These scenarios differ in (i) clean energy policies and (ii) regionalizations. In the left panel, we show results for State Policy, which means that each state has to meet its clean energy target (either RPS or CES, or both) in 2050. We list these targets in Table 1. In the right panel, we show results for the Regional 100% CES scenario enforcing that power generation in the WECC area comes from CES-eligible resources in 2050. Within the left panel, we show coordination scenarios of Incomplete Coordination, BAU (business-as-usual), Expanded EIM, Regional Market, and Full Coordination. Incomplete Coordination refers to the lowest level of coordination in the market, including limited transmission coordination. BAU represents the coordination as business-as-usual. Expanded EIM and Regional Market represent escalated coordinations on top of BAU. Full Coordination refers to the highest level of coordination, including enhanced generation coordination. Likewise, we show BAU, Expanded EIM, and Regional Market scenarios in the right panel – Regional 100% CES scenario. We revised the manuscript by adding a small reference to the

We made revisions throughout the manuscript to clarify scenario designs. For a quick review of which parameters are changed across scenarios, please see the Methods sections and the new Table 2 and Table 3.

- In Fig 2, why does the right panel not include a 0% in-state RPS and CES scenario like the one on the left?

0% in-state RPS and CES (now renamed to Full Coordination) would be redundant in the region-wide CES target (right panel in Figure 2). In the region-wide CES target, we enforce the model to have 100% CES over the entire WECC.

- Fig 3: what technologies for the non-emitting firm capacity are being built in the model?

In State Policy scenarios, natural gas combined cycle (non-CCS gas) resources are being built by the model. In Regional 100% CES scenarios, biopower, geothermal and natural gas combined cycle with carbon capture and storage (e.g., CCS gas) resources are being built by the model. We added a similar clarification around Figure 3 as follows: “While natural gas combined cycle resources are being built by the model in State

Policy scenarios, biopower, geothermal and CCS gas are being built by the model in Regional 100% CES scenarios.”

- I am surprised by the authors' conclusion that a 100% Regional CES scenario reduces exports across states, (p 12 line 239), is this relative to current state policy? I would expect higher levels of VRE to require more trade across the region to maintain balancing.

Indeed, this was a surprising finding as well. The key is this is relative to the State Policy scenario (see Figure 3) which achieves roughly 68% clean energy share, not a scenario without policy. We added the following parenthesis in the second paragraph of the “Transmission and storage complementarities and coordination” section: “...(relative to State Policy)...”

Our results also show that Regional 100% CES scenarios require significantly more VRE and storage, but it does not need as much transmission expansion as State Policy scenarios. Our interpretation is that states with low-ambition clean policy with high VRE potential export to other states with high-ambition policy in State Policy scenarios. However, the same low-ambition states keep this power in the state to meet its own target in a Regional 100% CES scenario. Thus, it reduces import/export and transmission expansion. We have a similar explanation in the Discussion and Conclusion section of the manuscript.

- What is “All” in Fig 5? Is that the scenario with existing clean energy policies?

Figure 5 shows the minimum capacities of storage, solar, and wind resources across scenarios. Bars with “100% CES” label in the x-axis show minimum capacities of solar, wind, and storage resources across BAU + Regional 100% CES, Expanded EIM + Regional 100% CES, and Regional Market + Regional 100% CES scenarios. Bars with an “All” label on the x-axis show minimum capacities of solar, wind, and storage resources across the main six scenarios (e.g., State Policy scenario is added to the combination above). We added a short explanation in the figure caption as follows: “All refers to the minimum capacity of a resource across the main six scenarios described in Figure 3 and Figure 4 and '100% CES' refers to the minimum capacity of a resource across three 100% CES scenarios.”

- I don't think the results in this section accurately reflect policy uncertainty. The authors state that going from “All” to a 100% CES scenario means that “policy uncertainty disappears” (p 12 line 258). I disagree, there is significant uncertainty behind whether a 100% CES would be implemented in the Western United States.

We agree in the sense that Regional 100% CES is just a possibility among many other future events. However, given that we have only two scenarios in the paper in terms of

clean energy policies (e.g., State Policy and Regional 100% CES policy), “change from All to 100% CES” refers to removing the impact of state policies. In other words, while making that inference, we assume that Regional 100% CES is the only possibility in the future.

Reviewer #3 (Remarks to the Author):

Summary

This paper deploys an optimization model that simulates the capacity expansion and coordination of electricity systems within the western U.S. The goal is to study the interaction of environmental goals (e.g. expansion of zero carbon electricity generation) with the planning and operation of the grid. The thought exercise is to ask how much more efficient would the delivery of varying levels of zero-carbon electricity be under different stylized representations of regional coordination.

General Comments

There is a lot of work that goes into building models like this so it is typical to see them deployed in multiple papers. Evaluating the quality and contribution of any single submission drawing upon the modeling framework then boils down to at least three criterion, A) how important is the problem they are deploying the model on?, B) how well does the modeling application represent the parameters of this interesting problem? and, C) how much can we believe the output of the model?

As a consumer, and occasional referee, of papers in this genre I constantly struggle with C. The problem is that there is no straightforward way to evaluate the quality of the optimization simulation models being deployed without really diving in and running it yourself, and I’m not able to do that in two weeks. Absent a full-on reproduction one would like to see some statistics or other evidence comparing model outputs to the real world. Unfortunately one can selectively choose which variables to highlight in such exercises, but I don’t see anything at all along those lines here.

We appreciate the sentiment as it is good practice to validate model results with real-world outcomes where available. As our modeling exercise focuses on a future year, 2050, this is not practical, and for other forward-looking energy modeling papers it is not common to attempt a validation (Long et al. 2021, Wei et al. 2019). Even near-term modeling studies (2030) do not typically approach validation due to the lack of real measures against which the validation would be performed (Energy Strategies, 2021b). We are able to compare some of our findings with other modeling studies and we note numbers produced to estimate historical Western EIM’s benefits (within the third paragraph of “Discussion and conclusion”). However, as we noted in the response to Reviewer 1, this is not an apples-to-apples comparison because our simulations are

run under different circumstances with different objectives. In the manuscript, we compare our results to other modeling studies' results wherever sufficient data are available from comparable modeling studies (e.g., storage capacity comparison with Long et al. 2021, transmission capacity comparison with Wei et al. 2019).

So going back to the first two criteria, I accept that the coordination, or lack thereof, of the operations and planning in power systems is an important question. My problem here is with the fact that the modeling approaches adopted here do not really capture the discrete dimensions of the "coordination" problem very well.

Thank you for recognizing the importance of the topic. We endeavor to capture as many discrete dimensions of coordination as possible, though there are some limitations. First, we are limited by modeling architecture and what can be implemented in an optimization framework. Non-linear aspects would make the model computationally intractable. Second, we are limited by available data to parameterize such coordination. For example, in an ideal world, we could determine all of the bilateral contracts made between states and set those as binding constraints. However, given these data are not available, hurdle rates are a commonly adopted solution.

For this paper specifically, the focus is to address the impact of institutional barriers in power systems on clean energy pathways. We do this by identifying the main sources of existing inefficiencies, and translating them into technical constraints or formulations in optimization problems. Given we cannot be exhaustive, we can say that the results of this study are an underestimation of the magnitude of the problem. We note this in the "Discussion and conclusion" of the revised manuscript as follows: "Our study does not capture all the discrete elements of the coordination due to computational tractability and data availability. Therefore, our results likely underestimate the benefits of coordination."

The challenge here is to use an optimization model to somehow represent how sub-optimal operations in given scenarios are. I think this is a question that can only be tackled by measurement of current outcomes that could be compared to hypothetical "optimal" ones. For some reason, this approach is more common to the economics literature than the OR/modeling literature. Even this is fraught with assumptions over what is actually optimal, or realistically achievable, but the approach used here is essentially to assume specific forms of "sub-optimal" within the context of the simulation model, and then study their removal. I do not find this convincing.

Econometrics approaches are useful for tackling "what-if" research questions retrospectively. As an example given in the comment below, Cicala (2022) assesses out-of-merit order dispatch due to inefficiencies such as balancing authority coordination and long-term contracting. This study, in fact, motivated our work. However,

econometrics and optimizations address fundamentally different research problems. Econometric approaches tackle research problems with a backward-looking approach. They require voluminous data and cannot easily project dynamics in energy markets, policy and infrastructure two decades from now. Optimization models, on the other hand, can be designed with forward-looking research questions in mind, given numerous modeling caveats outlined above. We address the literature on econometrics below and clarify in the revised manuscript (“Modeling inter-jurisdictional coordination”) how our study differs from econometrics studies.

I elaborate more on specific elements of this below.

1. As I understand it, dispatch coordination, or rather the lack of coordination is modeled here using hurdle rates. While this is standard practice for modeling approaches to this problem, I am not aware of empirical work demonstrating that it is a particularly effective or realistic way of trying to represent the existing lack of coordination. Therefore the model sets up a strawman in the form of hurdle rates and then removes it in the name of coordination. Other studies have done this, but I personally do not find that very convincing and more generally, it raises the question of what the contribution is here.

Much of the public discourse over whether the west should have an expanded ISO, expanded EIM or something else has overlooked, or at least struggled to come to grips with, the fact that there has already been a great degree of inter-state electricity trade in the west even in the absence of a formal market. Many believe, as do I, that a market-based ISO could probably do better than existing practice, but modelers have struggled to document the sources and magnitudes of the existing inefficiencies.

Indeed, hurdle rates are commonly used in modeling literature, thus we are not breaking new ground or claiming a novel contribution with their use. Since inter-state electricity trade outside of the EIM is usually with bilateral contracts, it is not possible to fully parameterize these and evaluate the exact sources of inefficiencies (and more importantly, how to model these inefficiencies). Due to these difficulties, as noted above, modeling studies tend to create a synthetic friction parameter in place such as the hurdle rate. We extend the existing literature by considering several other dimensions of coordination together, including: operating reserves, resource adequacy, and transmission and generation coordination (as an example of these studies in the context of U.S. see Energy Strategies, 2021b, and in context of Europe, see Bjørndal et al., 2018). We discuss how prior modeling studies approach these market frictions in the “Modeling inter-jurisdictional coordination” section of the original manuscript.

Indeed, my understanding is that the bulk of EIM benefits, at least in the early years (before 2016), came from expanding market-based dispatch software to new regions that had previously been balancing their systems using ad-hoc approaches. In other

words, it was less about coordination between regions and more about improving how each region handled its own internal resources in setting its “base schedules”. This may have shifted over time as EIM expanded and larger chunks of transmission rights were able to be deployed.

The point is, existing dispatch inefficiency, particularly in the day-ahead bilateral market in the west, are at best only crudely captured through imposition of hurdle rates.

Some of these elements are not going to be modeled explicitly. Hurdle rates are specifically designed to capture inefficiencies of bilateral contracting – the major of inter-state electricity trade in the west. Though, certainly, we agree that we cannot exhaustively capture all sources of inefficiency. We added a discussion stating the limitation of our study in the revised manuscript (see our response above about the discrete dimensions of the coordination).

To address the specific concern around hurdle rates, we run a suite of additional sensitivities. You can review our response to the Reviewer 1 comment above, supplementary information, and additional text in the revised manuscript. When we change hurdle rate costs, the system-wide cost is changed by less than 1% of the BAU scenario. Overall resource capacities do not change significantly in the region, but geographic distribution of the resources changes. The most sensitive resources to hurdle rates are wind, solar, and non-CCS gas. These resources change their investment locations from modeling zones with higher hurdle rate costs to those with lower costs. The overall system cost does not change significantly, but cost components change. With lower hurdle rate costs, total transaction cost decreases, while costs arising from transmission expansion grow.

2. Reserve coordination is modeled by allowing reserves to be spread over concentrically larger geographic areas under different scenarios of coordination. Again, I don’t think this is a particularly accurate representation of how reserves are implemented under the EIM, or how much potential there could be with full market integration.
 - a. First, my understanding is that reserves are not co-optimized in the EIM (at least outside of CAISO) and each jurisdiction has to self provide and manage its own reserves. The only tangible reserve benefit to regionalization comes from a technical “diversity benefit“ that is granted to regions toward passing a resource sufficiency evaluation that is a condition of EIM participation.

Indeed that is our understanding of the current reserve treatment in the EIM, which we noted above in response to a question by Reviewer 1 about our reserve formulation. We include reserve sharing as a dimension of regionalization scenarios because it has been considered an important element of market coordination. Our results show that including

reserve sharing leads to a \$250M annual benefit when increasing the coordination from BAU to Expanded EIM (after removing the effect of hurdle rates). In fact, most existing capacity expansion models for the region neglect operating reserves entirely (Long et al., 2021).

- b. Second, there are real limits as to how far reserves can be spread given the physical limitations of the grid, and there is some concern that reserves, which are usually procured at a balancing authority level, are already unable to be delivered to the places where they will really be needed. In other words, there is pressure to make at least some reserves more local, not less local. Thus, there is a distinction between coordination and geographic scope.

It is correct that we do not have deliverability details in our models, besides the inclusion of long-term contractual imports/exports (described above in response to Reviewer 1). This is one of the limitations of our study. Adding deliverability details would substantially complicate the model and require collecting additional reserve data, which may not be public at this time. We indicated this limitation in the “Discussion and conclusion” of the revised manuscript as follows: “Operating reserves are shared over the region when coordination increases, but we do not have deliverability details other than long-term contractual imports and exports.”

3. It wasn't fully clear to me how resource adequacy (or planning) is modeled, with or without coordination. It again appears to be a matter of geographic scope. However, it's worth pointing out that even under current RA policies, the trade of capacity between balancing authorities is allowed. Therefore a representation of BAU that assumes all capacity needed to meet RA goals must be local would be incorrect. Conversely, I'm not sure what the best way to model “coordinated” resource adequacy policy would be, as the mechanisms that link any specific RA policy and investment choices have not been empirically demonstrated.

We show in Table 2 of the revised manuscript that resource adequacy is maintained at the zonal level in the BAU and the Expanded EIM scenarios, and region-wide in the Regional Market scenario. We elaborated the Method section with further explanation in this line.

Even in the BAU scenario we allow for fixed import/export contributions in resource adequacy planning as tabulated by California Energy Commission (2020) – a report that California Energy Commission, California Public Utilities Commission, and California Air Resources Board should submit to the Legislature every four years – and the WECC. This type of modeling allows balancing authorities (in our case, modeling zones) to cover a part of resource adequacy via out-of-state resources. Hence, we do not claim that all resources meeting resource adequacy goals should be in-state in BAU. We do

not model resource adequacy from a policy-investment perspective and we are not coordinating investors and regulators. Rather, we are coordinating how states plan to meet their resource adequacy requirements.

4. With regards to transmission, the paper seems to use the term “coordination” as a catch-all term for the absence of a wide variety of political and economic barriers. This is best illustrated by the description on page 22, “we allow the model to build along any feasible pathway without any maximum capacity limit.” The coordination of transmission investment therefore seems to involve the removal not just of the frictions created by inter-jurisdictional decision making but also of lawsuits, environmental regulations such as California’s CEQA, and general NIMBYism, that would exist under most any plausible coordination scenario.

We would like to clarify that Transmission Coordination in transmission planning refers to “...build along any feasible pathway without any maximum capacity limit...”. Feasible pathways refer to current existing transmission pathways between modeling zones. Limited Transmission Coordination limits transmission expansion to only those pathways which have completed some significant level of pre-planning as of today (we call these projects “proposed projects” in the manuscript). Given the challenges of building new transmission lines, we think it is a useful bounding exercise to only include the option to choose among those projects which have undergone some level of vetting and stakeholder consultation. We give a list of these projects in Supplementary Table 15.

We are aware that even the enhanced coordination of transmission investment should/would include many elements of friction, including policies, legal actions, environmental concerns, cost-sharing objections, etc. We are aware and cite modeling literature whose sole focus is to analyze the impact of environmental regulations on renewable project siting (Wu et al., 2020). Additionally, there are other studies that analyze the benefit of coordination in transmission expansion on the whole system cost, but they ignore environmental siting constraints (Brown and Botterud, 2021, Brinkman et al., 2021). We think the inclusion of such barriers in our model should be the scope of future extensions of this work.

5. On line 254 the authors’ state that “Robust decisions reflect no-regrets investment regardless of policy uncertainties” and note that there is a doubling of “no-regrets capacity when policy uncertainty disappears”. This would be more informative if the authors could provide a precise definition of what they mean by “no-regrets”.

We put an explanation of “no-regrets” in the revised manuscript. It follows in the first paragraph of that section: “...no-regret capacity refers to the capacity that is built

regardless of which scenario is realized and thus represents a robust decision by states and utilities to future uncertainties...”

6. A minor quibble on the title. This study seems to be less about policy “uncertainty” and more about bad (or less bad) policies. It wasn’t clear where the uncertainty came in.

Allow us to justify the use of the word “uncertainty” in the title. We build a capacity expansion model that looks into future climate targets and possible scenarios, which incorporate many levels of uncertainty. One climate mitigation scenario, State Policy, envisions that each state meets its own target in 2050. The other set of scenarios relate to market regionalization, e.g., Regional 100% CES envisions that each state in WECC collaboratively meets the 100% CES target in 2050. Optimal design of power systems under these scenarios are provided by the model.

We discuss the importance of parametric and structural uncertainties in energy system modeling in the original manuscript.

7. The paper is long on citations from industry and government reports and short on citations from the academic literature. There is an optimization model based literature and the more econometric based economics literature that both study the efficiency of power market operations that exist in almost parallel universes. The extensive work that has appeared in economics journals is missing. This may be fine if this were an operations research journal (I actually don’t think so), but is a problem for a general journal such as this one. I provide a partial list below.

The single most glaring omission is Cicala (2022) who studies the kinds of questions being asked here, only retrospectively, by using the staggered timing of state deregulation and ISO expansions to test for changes in his measure of dispatch and trade efficiency. Another relevant study that is not that well known is the working paper by Mansur and White who study the event of PJM’s expansion in to Ohio.

Thank you for your suggestions. We have reworked the literature review to give better context for econometrics studies (elaborated in response to a comment above). We added the following paragraph in the revised manuscript: “The efficiency of electricity markets has been evaluated by econometrics models retrospectively based on empirical data. The benefit of market and different dispatch schemes on generation output and cost were examined using a difference-in-difference analysis considering the staggered transition to markets (Cicala, 2022). The benefit of centralized dispatch in the context of PJM’s expansion to areas previously trading electricity with bilateral contracts has been shown to be large (Mansur and White, 2012). These results follow from a long literature noting the prospective benefits of greater market integration (Davis and Wolfram, 2012; Fabrizio et al, 2007; Joskow, 1997, 2012). Econometric techniques are

well-suited when there is abundant data and the research focus can be a well-specified set of historical events. In order to assess forward-looking outcomes, e.g., for 100% clean energy by mid-century, then a different class of models is needed, such as optimizations which can consider dynamics of changing markets, policies and infrastructures while respecting technical constraints.”

References

- Cicala, Steve. 2022. Imperfect Markets versus Imperfect Regulation in US Electricity Generation. *American Economic Review*, 112 (2): 409-41.
- Davis LW, Wolfram C. 2012. Deregulation, Consolidation, and Efficiency: Evidence from U.S. Nuclear Power. *American Economic Journal: Applied Economics*. 4:194- 225.
- Fabrizio K, Rose NL, Wolfram C. 2007. Do markets reduce costs? Assessing the impact of regulatory restructuring on U.S. electric generation efficiency. *American Economic Review*. 97:1250-1277.
- Joskow PL. 1997. Restructuring, competition and regulatory reform in the U.S. electricity sector. *Journal of Economic Perspectives* 11:119-138.
- Joskow PL. 2006. Markets for power in the United States: An interim assessment. *The Energy Journal* 27: 1-36.
- Joskow PL. 2012. Creating a smarter U.S. electricity grid. *Journal of Economic Perspectives*. 26: 29-48.
- Mansur ET, White MW. 2012. Market Organization and Efficiency in Electricity Markets. Dartmouth University Working Paper. [http : //www.dartmouth.edu/ mansur/papers/m](http://www.dartmouth.edu/mansur/papers/m)

Our references

- Bjørndal, E., Bjørndal, M., Cai, H. and Panos, E. (2018) Hybrid pricing in a coupled European power market with more wind power. *European Journal of Operational Research*, 264(3), 919-931. <https://doi.org/10.1016/j.ejor.2017.06.048>
- Brinkman G., Bain D. and Buster G. (2021) The North American renewable integration study (NARIS): A U.S. perspective. Tech. Rep. NREL/TP-6A20-79224, 1804701, MainId:33450, National Renewable Energy Laboratory, <https://doi.org/10.2172/1804701>, URL <https://www.osti.gov/servlets/purl/1804701/>

Brown, P. R. and Botterud, A. (2021) The value of inter-regional coordination and transmission in decarbonizing the US electricity system. *Joule*, 5(1), 115–134.
<https://doi.org/10.1016/j.joule.2020.11.013>

California Energy Commission (2020) Input & Assumptions - CEC SB 100 Joint Agency Report. URL
<https://efiling.energy.ca.gov/GetDocument.aspx?tn=234532&DocumentContentId=67359>

California Public Utilities Commission (2023) System Reliability Modeling Datasets 2023. Accessed October 16, 2024.
www.cpuc.ca.gov/industries-and-topics/electrical-energy/electric-power-procurement/long-term-procurement-planning/2022-irp-cycle-events-and-materials/system-reliability-modeling-datasets-2023

Catchpole, D. (2024) Northwest reserve-sharing group now covers almost all of Western Interconnection. NewsData, LLC, May 31, 2024.
https://www.newsdata.com/clearing_up/briefs/northwest-reserve-sharing-group-now-covers-almost-all-of-western-interconnection/article_1216f554-1eb0-11ef-bbf5-c3ab9815f379.html.

Cicala, S. (2022) Imperfect markets versus imperfect regulation in US electricity generation. *American Economic Review*, 112 (2) 409-441.
<https://doi.org/10.1257/aer.20172034>

Clean Energy States Alliance (2024) Table of 100% clean energy states. Accessed October 15, 2024.
<https://www.cesa.org/projects/100-clean-energy-collaborative/guide/table-of-100-clean-energy-states/>.

Davidson, M. R., Filatova, T., Peng, W., Verbeek, L. and Kucuksayacigil, F. (2024) Simulating institutional heterogeneity in sustainability science. *Proceedings of the National Academy of Sciences*, 121(8), e2215674121
<https://doi.org/10.1073/pnas.2215674121>.

DiFabio, C. (2024) New members successfully integrated into Northwest Power Pool Reserve Sharing Group. Western Power Pool, May 28, 2024.
<https://www.westernpowerpool.org/news/new-members-successfully-integrated-into-northwest>.

Energy Strategies (2021a) The state-led market study - Exploring western organized market configurations: A western states' study of coordinated market options to advance state energy policies. (Market and Regulatory Review Report) Tech. Rep. Energy Strategies

<https://static1.squarespace.com/static/59b97b188fd4d2645224448b/t/6148a03ea5c43d63b2873506/1632149569046/Final+Roadmap+-+Market+and+Regulatory+Review+Report+210730.pdf>

Energy Strategies (2021b) The state-led market study - Exploring western organized market configurations: A western states' study of coordinated market options to advance state energy policies. (Technical Report) Tech. Rep. Energy Strategies

<https://static1.squarespace.com/static/59b97b188fd4d2645224448b/t/6148a012aa210300cbc4b863/1632149526416/Final+Roadmap+-+Technical+Report+210730.pdf>

Johnston, J., Henriquez-Auba, R., Maluenda, B. and Fripp, M. (2019) Switch 2.0: A modern platform for planning high-renewable power systems. *SoftwareX*, 10, 100251.

<https://doi.org/10.1016/j.softx.2019.100251>

Lew, D., Brinkman, G., Ibanez, E., Florita, A., Heaney, M., Hodge, B.-M., Hummon, M., Stark, G., King, J., Lefton, S. A., Kumar, N., Agan, D., Jordan, G. and Venkataraman, S. (2013) The western wind and solar integration study phase 2. Tech. Rep.

NREL/TP-5500-55588, National Renewable Energy Laboratory, URL

<https://doi.org/10.2172/1220243>

Long, J. C. S., Baik, E., Jenkins, J. D., Kolster, C. L. E. A., Chawla, K., Olson, A. R. N. E., Cohen, A., Colvin, M., Benson, S. M. and Jackson, R. B. (2021) Clean firm power is the key to California's carbon-free energy future. *Issues in Science and Technology*.

<https://issues.org/california-decarbonizing-power-wind-solar-nuclear-gas/>

Munoz, F. D., Sauma, E. E. and Hobbs, B. F. (2013) Approximations in power transmission planning: Implications for the cost and performance of renewable portfolio standards. *Journal of Regulatory Economics*, 43(3), 305–338.

<https://doi.org/10.1007/s11149-013-9209-8>

Munoz, F. D., Van Der Weijde, A. H., Hobbs, B. F., and Watson, J.-P. (2017). Does risk aversion affect transmission and generation planning? A Western North America case study. *Energy Economics*, 64, 213–225. <https://doi.org/10.1016/j.eneco.2017.03.025>

National Caucus of Environmental Legislators (2023) Colorado sets targets to eliminate greenhouse gas emissions by 2050. Accessed October 15, 2024.

<https://www.ncelenviro.org/articles/colorado-sets-targets-to-eliminate-greenhouse-gas-emissions-by-2050/>.

National Economic Council (2000) Taking action to ensure the federal government does its part to help California meet its electricity needs. Accessed October 16, 2024

[https://clintonwhitehouse4.archives.gov/WH/EOP/nec/html/doc080300.html#:~:text=The%20President%20emphasized%20that%20as%20one%20of,2%25%20of%20total%20electricity%20use\)%2C%20the%20Federal](https://clintonwhitehouse4.archives.gov/WH/EOP/nec/html/doc080300.html#:~:text=The%20President%20emphasized%20that%20as%20one%20of,2%25%20of%20total%20electricity%20use)%2C%20the%20Federal)

National Renewable Energy Laboratory (2017) Western wind and solar integration study. Tech. Rep. NREL/SR-550-47434, Grid Modernization, URL

<https://www.nrel.gov/grid/wwsis.html>

Perez, A. P., Sauma, E. E., Munoz, F. D., and Hobbs, B. F. (2016) The economic effects of interregional trading of renewable energy certificates in the U. S. WECC. *The Energy Journal*, 37(4), 267–296. <https://doi.org/10.5547/01956574.37.4.aper>

Sergi, B. and Cole, W. (2021) Operating reserves in ReEDS. National Renewable Energy Laboratory, URL www.nrel.gov/docs/fy22osti/81706.pdf

Stephen, G. (2021) Probabilistic resource adequacy suite (PRAS) v0.6 model documentation. Tech. Rep. NREL/TP-5C00-79698, National Renewable Energy Laboratory, URL <https://www.nrel.gov/docs/fy21osti/79698.pdf>

Wei, M., Raghavan, S. V. and Hidalgo-Gonzalez, P. (2019) Building a healthier and more robust future: 2050 low-carbon energy scenarios for California. California Energy Commission.

<https://www.energy.ca.gov/publications/2019/building-healthier-and-more-robust-future-2050-low-carbon-energy-scenarios>

Wu, G. C., Leslie, E., Sawyerr, O., Cameron, D. R., Brand, E., Cohen, B., Allen, D., Ochoa, M. and Olson, A. (2020) Low-impact land use pathways to deep decarbonization of electricity. *Environmental Research Letters*, 15(7), 074044.

<https://doi.org/10.1088/1748-9326/ab87d1>

Response to Reviewers:

“Coordinating Power Sector Climate Transitions Under Policy Uncertainty”

Fikri Kucuksayacigil, Zhenhua Zhang and Michael R. Davidson

February 13, 2025

Legend:

Black text – Reviewer comments

Red text – Response to reviewers

Blue text – New text added to paper

REVIEWER COMMENTS

Reviewer #1 (Remarks to the Author):

Thank you for the detailed responses to the reviewer comments. I feel like the manuscript is much improved from the original submission. A couple notes on the responses:

Thank you for your comments.

- With respect to the operating reserve sharing, I was referring to the shared requirement for the reserves (I recognize that wasn't clear). If each utility set its operating reserve requirement based on being an islanded system, the total requirements would be much higher. So there is already some intrinsic value in coordination/sharing that is part of the system. No need to address this in the manuscript, just trying to clarify the comment.

- I really like the addition of the sensitivity scenarios. I feel like that helps put your results in better context, and helps the reader understand the potential impact of the assumptions you have made along the way.

- I appreciate the pointers to the code and SI. I think I missed many of these the first time around simply because they weren't easy for me to find.

Thank you for the clarification and additional comments.

My biggest concern with the paper is still that the baseline scenario that you are measuring against is too coordinated relative to real life, largely because of the large-ish region sizes you are using. There is still plenty of "within-region" friction that is present today that cannot be captured by this model at its current spatial resolution. In your discussion section on page 19 I recommend making your discussion of the limitation of in-state friction a little more clear, perhaps by noting that all western states have more than one utility, so in-state frictions between those utilities are not captured in the scenarios modeled.

As recommended, we revised the Discussion section. The following is the whole paragraph with the revision in bold: “Our study has several limitations. We model coordination through five distinct functions, though in reality, there are many stages through to full regionalization. For instance, instead of joining a full market, some states and utilities may be more prepared to join real-time markets, day-ahead markets, and/or regional resource adequacy programs. It is also possible that states and utilities will choose different configurations---including multiple regionalization entities (such as bifurcated EIMs), a regional market without key states, or where utilities in the same state join different RTOs (Energy Strategies, 2021, Siemens Industry, 2021). Our study does not capture all the discrete elements of coordination due to computational tractability and data availability, thereby underestimating the benefits of coordination. Our study does not incorporate transmission network expansion that must take place within states for feasible dispatch. What determines states' RTO participation is an open area of debate: policy diffusion could lead to a domino effect of states choosing regional coordination. Our modeling results rely on sampled weeks (including peak week) from a single year, **ignoring potential extreme conditions. Furthermore, we utilize aggregated modeling zones that ignore in-state frictions. For example, all western states have more than one utility, and power transfers between those in-state utilities are subject to friction, ignored here. Higher spatial granularity would come with important computational and practical challenges.** Operating reserves are shared over the region when coordination increases, but we do not have deliverability details other than long-term contractual imports and exports. We believe these provide promising directions for future research.”

Reviewer #2 (Remarks to the Author):

I reviewed the authors' revisions in response to my report and am OK with how they have addressed my peer review.

Thank you for your comments.

Referee Report: Revision of “Coordinating Power Sector Climate Transitions Under Policy Uncertainty”

Summary

This paper deploys an optimization model that simulates the capacity expansion and coordination of electricity systems within the western U.S. The goal is to study the interaction of environmental goals (e.g. expansion of zero carbon electricity generation) with the planning and operation of the grid. The thought exercise is to ask how much more efficient would the delivery of varying levels of zero-carbon electricity be under different stylized representations of regional coordination.

General Comments

There is a lot of work that goes into building models like this so it is typical to see them deployed in multiple papers. Evaluating the quality and contribution of any single submission drawing upon the modeling framework then boils down to at least three criterion, A) how important is the problem they are deploying the model on?, B) how well does the modeling application represent the parameters of this interesting problem? and, C) how much can we believe the output of the model?

As a consumer, and occasional referee, of papers in this genre I constantly struggle with C. The problem is that there is no straightforward way to evaluate the quality of the optimization simulation models being deployed without really diving in and running it yourself, and I’m not able to do that in two weeks. Absent a full-on reproduction one would like to see some statistics or other evidence comparing model outputs to the real world. Unfortunately one can selectively choose which variables to highlight in such exercises, but I don’t see anything at all along those lines here.

So going back to the first two criterion, I accept that the coordination, or lack thereof, of the operations and planning in power systems is an important question. My problem here is with the fact that the modeling approaches adopted here to do not really capture the discrete dimensions of the “coordination” problem very well.

The challenge here is to use an optimization model to somehow represent how *sub-optimal* operations in given scenarios are. I think this is a question that can only be tackled by measurement of current outcomes that could be compared to hypothetical “optimal” ones. For some reason, this approach is more common to the economics literature than the OR/modeling literature. Even this is fraught with assumptions over what is actually optimal, or realistically achievable, but the approach used here is essentially to assume specific forms of “sub-optimal” within the context of the simulation model, and then study their removal. I do not find this convincing.

I elaborate more on specific elements of this below.

1. As I understand it, dispatch coordination, or rather the lack of coordination is modeled here using hurdle rates. While this is standard practice for modeling approaches to this problem, I am not aware of empirical work demonstrating that it is a particularly effective or realistic way of trying to represent the existing lack of coordination. Therefore the model sets up a strawman in the form of hurdle rates and then removes it in the name of coordination. Other studies have done this, but I personally do not find that very convincing and more generally, it raises the question of what the contribution is here.

Much of the public discourse over whether the west should have an expanded ISO, expanded EIM or something else has overlooked, or at least struggled to come to grips with, the fact that there has already been a great degree of inter-state electricity trade in the west even in the absence of a formal market. Many believe, as do I, that a market-based ISO could probably do better than existing practice, but modelers have struggled to document the sources and magnitudes of the existing inefficiencies.

Indeed, my understanding is that the bulk of EIM benefits, at least in the early years (before 2016), came from expanding market-based dispatch software to new regions that had previously been balancing their systems using ad-hoc approaches. In other words, it was less about coordination between regions and more about improving how each region handled its own internal resources in setting its “base schedules”. This may have shifted over time as EIM expanded and larger chunks of transmission rights were able to be deployed.

The point is, existing dispatch inefficiency, particularly in the day-ahead bilateral market in the west, are at best only crudely captured through imposition of hurdle rates.

2. reserve coordination is modeled by allowing reserves to be spread over concentrically larger geographic areas under different scenarios of coordination. Again, I don’t think this is a particularly accurate representation of how reserves are implemented under the EIM, or how much potential there could be with full market integration.
 - First, my understanding is that reserves are not co-optimized in the EIM (at least outside of CAISO) and each jurisdiction has to self provide and manage its own reserves. The only tangible reserve benefit to regionalization comes from a technical “diversity benefit“ that is granted to regions toward passing a resource sufficiency evaluation that is condition of EIM participation.
 - Second, there are real limits as to how far reserves can be spread given the physical limitations of the grid, and there is some concern that reserves, which are usually procured at a balancing authority level, are already unable to be

delivered to the places where they will really be needed. In other words, there is pressure to make at least some reserves *more* local, not less local. Thus, there is a distinction between coordination and geographic scope.

3. It wasn't fully clear to me how resource adequacy (or planning) is modeled, with or without coordination. It again appears to be a matter of geographic scope. However, it's worth pointing out that even under current RA policies, the trade of capacity between balancing authorities is allowed. Therefore a representation of BAU that assumes all capacity needed to meet RA goals must be local would be incorrect. Conversely, I'm not sure what the best way to model "coordinated" resource adequacy policy would be, as the mechanisms that link any specific RA policy and investment choices have not been empirically demonstrated.
4. With regards to transmission, the paper seems to use the term "coordination" as a catch all term for the absence of a wide variety of political and economic barriers. This is best illustrated by the description on page 22, "we allow the model to build along any feasible pathway without any maximum capacity limit." The coordination of transmission investment therefore seems to involve the removal not just of the frictions created by inter-jurisdictional decision making but also of lawsuits, environmental regulations such as California's CEQA, and general NIMBYism, that would exist under most any plausible coordination scenario.
5. On line 254 the authors' state that "Robust decisions reflect no-regrets investment regardless of policy uncertainties" and note that there is a doubling of "no-regrets capacity when policy uncertainty disappears". This would be more informative if the authors could provide a precise definition of what they mean by "no-regrets".
6. A minor quibble on the title. This study seems to be less about policy "uncertainty" and more about bad (or less bad) policies. It wasn't clear where the uncertainty came in.
7. The paper is long on citations from industry and government reports and short on citations from the academic literature. There is an optimization model based literature and the more econometric based economics literature that both study the efficiency of power market operations that exist in almost parallel universes. The extensive work that has appeared in economics journals is missing. This may be fine if this were an operations research journal (I actually don't think so), but is a problem for a general journal such as this one. I provide a partial list below.

The single most glaring omission is Cicala (2022) who studies the kinds of questions being asked here, only retrospectively, by using the staggered timing of state deregulation and ISO expansions to test for changes in his measure of dispatch and

trade efficiency. Another relevant study that is not that well known is the working paper by Mansur and White who study the event of PJM's expansion in to Ohio.

1 References

Cicala, Steve. 2022. Imperfect Markets versus Imperfect Regulation in US Electricity Generation. *American Economic Review*, 112 (2): 409-41.

Davis LW, Wolfram C. 2012. Deregulation, Consolidation, and Efficiency: Evidence from U.S. Nuclear Power. *American Economic Journal: Applied Economics*. 4:194-225.

Fabrizio K, Rose NL, Wolfram C. 2007. Do markets reduce costs? Assessing the impact of regulatory restructuring on U.S. electric generation efficiency. *American Economic Review*. 97:1250-1277.

Joskow PL. 1997. Restructuring, competition and regulatory reform in the U.S. electricity sector. *Journal of Economic Perspectives* 11:119-138.

Joskow PL. 2006. Markets for power in the United States: An interim assessment. *The Energy Journal* 27: 1-36.

Joskow PL. 2012. Creating a smarter U.S. electricity grid. *Journal of Economic Perspectives*. 26: 29-48.

Mansur ET, White MW. 2012. Market Organization and Efficiency in Electricity Markets. Dartmouth University Working Paper. <http://www.dartmouth.edu/mansur/papers/r>